# Testing Seismic Amplitude Source Location (ASL) for Fast Debris Flow Detection at Illgraben, Switzerland

Fabian Walter[1,2], Arnaud Burtin[3,4], Brian W. McArdell[2], Niels Hovius[3], Bianca Weder[1] and Jens M. Turowski[3]

[1]Laboratory of Hydraulics, Hydrology and Glaciology, ETH Zurich, 8093, Switzerland
[2]Swiss Federal Institute for Forest, Snow and Landscape Research WSL, 8903, Switzerland
[3]GFZ German Research Centre for Geosciences, 14473, Germany
[4]Institut de Physique Du Globe de Paris, 75238, France

*Correspondence to*: Fabian Walter (walter@vaw.baug.ethz.ch)

**Abstract.** Heavy precipitation can mobilize tens to hundreds of thousands of cubic meters of sediments in steep Alpine torrents in a short time. The resulting debris flows (mixtures of water, sediments and boulders) move downstream with velocities of several meters per second and have a high destructive potential. Warning protocols for affected communities rely on raising awareness to the debris flow threat, precipitation monitoring and rapid detection methods. The latter, in particular, is a challenge, because debris-flow-prone torrents have their catchments in steep and inaccessible terrain, where instrumentation is difficult to install and maintain. Here we test amplitude source location (ASL) as a processing scheme for seismic network data for early warning purposes. We use debris flow and noise seismograms from the Illgraben catchment, Switzerland, a torrent system, which produces several debris flow events per year. Automatic in-situ detection is currently based on geophones mounted on concrete check dams and radar stage sensors suspended above the channel. The ASL approach has the advantage that it uses seismometers, which can be installed at more accessible locations, and where a stable connection to mobile phone networks is available for data communication. Our ASL processing uses time-averaged ground vibration amplitudes to estimate the location of the debris flow front. Applied to continuous data streams, inversion of the seismic amplitude decay throughout the network is robust and efficient, requires no manual identification of seismic phase arrivals and eliminates the need for a local seismic velocity model. We apply the ASL technique to a small debris flow event on 19 July 2011, which was captured with a temporary seismic monitoring network. The processing rapidly detects the debris flow event half an hour before arrival at the outlet of the torrent and several minutes before detection by the in situ alarm system. An analysis of continuous seismic records furthermore indicates that detectability of Illgraben debris flows of this size is unaffected by changing environmental and anthropogenic seismic noise and that false detections can be greatly reduced with simple processing steps.

# 1 Introduction

Debris flows threaten human lives and infrastructure in Alpine regions, including Switzerland (e.g. Badoux et al., 2016; Hilker et al., 2009). Real-time monitoring and forecasts of rainfall can identify exceedance of a precipitation threshold beyond which debris flows are likely triggered (e.g., Wieczorek 1987; Deganutti et al. 2000; Fan et al. 2003). Such alarms are useful to raise the general level of alert, but often not accurate enough to serve as a basis for rescue deployment, road closure or building evacuation. Furthermore, empirical relationships between rainfall and debris flow initiation are not necessarily transferable to other regions, because the hydrological response of a catchment depends on the amount of precipitation (Gregoretti et al., 2016) and may react to sudden environmental changes such as wildfires (Cannon et al., 2008; Kean et al., 2012; Rengers et al., 2016).

What further complicates precipitation-based alarms is that other sources of water may also be involved, such as snowmelt, and hence the relation between precipitation and debris flow initiation is complex. Initial sediment mobilization can either be triggered via increased ground pore water pressures leading to failure across a critical subsurface layer ("landslide triggering") or water drag forces of surface runoff (Berti and Simoni, 2005 and references therein; Godt and Coe, 2007). This initial mobilization may occur on lateral slopes or within the torrent channel (Gregoretti and Fontana, 2008). Debris flow triggering is also possible in the absence of precipitation, when natural lake dams formed by landslides or glacial ice, for example, suddenly rupture and subsequently a critical runoff results (Costa and Schuster, 1988; Evans and Clague, 1994). This complexity of triggering processes suggests that for warning purposes, rapid detection of debris flow formation may be more appropriate than rain forecasting.

A variety of instruments have been developed for rapid debris flow detection (for an instrumentation review, see Arattano and Marchi, 2008). Certain instruments such as trip wires and pendulums require direct contact with the debris flow and possibly replacement after an event. Moreover, these devices are notoriously prone to false detection. The direct-contact requirement does not apply to ultrasonic, radar and laser altimeters for flow depth measurements. However, in any case, devices have to be suspended above the torrent bed and therefore require stable banks, a condition, which often is not met. Human observers can provide reliable detections of debris flows, but this approach is often not feasible in remote terrain (Marcial et al., 1996).

Seismological techniques constitute another approach to tackle the challenge of reliable debris flow detection. Alpine mass movements often involve processes that generate seismic waves detectable at kilometer distances (Burtin et al., 2016). Ground impact of rock falls (Deparis et al., 2008), particle hopping during bedload transport in rivers (Burtin et al., 2008; Tsai et al., 2012; Gimbert et al., 2014) and snow avalanche or landslide interaction with obstacles (Surinach et al., 2000, Dammeier et al., 2011) all transmit high frequency (>1 Hz) seismic energy to the ground. Consequently, with the advent of

more portable sensor and recorder technology, seismology has become increasingly popular in natural hazard and in particular debris flow research (Galgaro et al., 2005; LaHusen et al., 2005; Huag et al., 2007; Cole et al., 2009; Zobin et al., 2009; Abancó et al., 2012; Vázquez et al., 2016). Unlike landslides, avalanches and rock falls, debris flows typically move at relatively slow velocities below 10 m/s (e.g. Hürlimann et al., 2003). In principle, seismic monitoring thus allows for considerable warning time provided that detection occurs rapidly upon debris flow initiation.

Exploiting seismology for early warning requires both rapid detection and location of the debris flow front. One approach is to place sensors as close to the torrent channel as possible and to monitor the ground vibration amplitude as the debris flow front passes the sensor. For such set ups, ground motions of up to $2\times10^{-3}$ m/s (Hübl et al., 2013) have been observed, covering a frequency spectrum between a few and several hundred Hz (Burtin et al., 2014; Marcial et al., 1996; Lavigne et al., 2000). Detection of the debris flow front is thus possible tens of seconds before its arrival, which can be improved with the additional sensing of sound waves traveling through air (Arattano, 1999; Arratano and Marchi, 2005; Hübl et al., 2013; Schimmel and Hübl, 2015). If ground vibration data is efficiently transmitted and monitored remotely, such near-torrent installations can result in detections of debris flow fronts up to an hour before they move into inhabited areas (Marcial et al., 1996).

Seismic sensors can also be placed behind steel plates installed flush with the torrent bed. Due to their sensitivity to higher frequencies (> 1 Hz), the seismic sensors used in this setup are typically referred to as geophones rather than seismometers. Ground motion sensing is almost exclusively confined to sediments moving directly across the steel plate. This has become an attractive method to monitor bedload transport (e.g. Turowski et al., 2015; Wyss et al., 2016) and can increase the detection and location accuracy of the debris flow front (Badoux et al., 2009). However, such in situ installations are technically more challenging and sediment accumulation above the steel plate often compromises detection.

Despite its success in debris flow detection, seismic installation near and within torrents is not ideal, because instruments may be damaged by rock falls or the debris flows they are supposed to monitor. Moreover, torrents with steep canyon walls are often shielded from mobile phone networks and direct sunlight needed for real-time data communication and solar power supply. These problems can be overcome by placing seismometers further away from the torrent, which also allows some control on structural site effects (Hürlimann et al., 2003).

When focusing on frequencies of a few Hz rather than the 10's to 100's of Hz used for near-torrent monitoring, signal attenuation is mitigated and sensors at distances of 100's or 1000's of meters from the torrent can still detect debris flows (Burtin et al., 2016). However, seismic source location, which is necessary to distinguish debris flow fronts from other ambient seismic sources (Schimmel and Hübl, 2015; Burtin et al., 2014; Arattano et al., 2014), is more challenging when sensors are separated from the torrent. Lack of clear seismic phase arrivals and signal coherence throughout seismic

networks covering entire catchments prohibit the use of traditional seismic source location based on arrival time inversion (e.g. Diehl et al., 2009) and array techniques (e.g. Rost and Thomas, 2002).

Alternatively, seismic source locations can be obtained with seismogram amplitude information. In this way, the location of a debris flow or its front can be determined by identifying the point in space, which best models the amplitudes of debris flow seismograms recorded throughout a network. The technique is often referred to as Amplitude Source Location (ASL) and has been applied to locating different kinds of mass motion including debris flows (Yamasato 1997; Kumagai, 2009; Walsh et al., 2016; Jolly et al., 2002; Aki and Ferrazzini, 2000; Ogiso and Yomogida, 2015) and to other seismic events (Battaglia and Aki, 2003; Ogiso and Yomogida, 2012) in volcanic regions as well as to seismic sources in glaciated environments (Jones et al., 2013; Röösli et al., 2014).

The variety of applications of the ASL technique testifies to its flexibility. Unlike arrival time-based location, ASL does not require knowledge of a seismic velocity model. Geological and topographical heterogeneity can often be neglected or parameterized as site response using earthquake data (e.g. Kumagai et al. 2009) or artificial seismic sources (Walsh et al., 2016). The robustness and straightforward implementation of the ASL technique has led various authors to suggest that it could be used in automated early warning and hazard mitigation schemes (Kumagai et al. 2009; Ogiso and Yomogida 2015; Jolly et al., 2002). Using real-time data communication via satellite and portable phone networks, this possibility is becoming more and more realistic. It remains to be shown, however, if the ASL technique is reliable enough to replace or supplement near-torrent and in-torrent installations. This point is the motivation of the present study.

Here, we explore the suitability of the ASL technique for early warning against debris flows at the Illgraben catchment, Switzerland. The Illgraben torrent produces several debris flows per year and is subject to state-of-the-art geophone plates and flow-depth altimeters, which are used for rapid debris flow detection and warning purposes (Badoux et al., 2009). To test the ASL method, we use archived seismic data from a debris flow event on 19 July 2011 recorded with a 10-station network (Burtin et al., 2014). The ASL technique identifies the formation of the debris flow front high up in the Illgraben catchment, where no in-torrent and near-torrent instrumentation is feasible. The results indicate that our approach is suitable for typical seismic records of debris flows, which do not include extensive signals from lateral slope erosion.

## 2 Illgraben Debris Flows

The Illgraben drains a catchment of 10 km$^2$ (Fig. 1) and transports large amounts of sediment to the Rhone River, as is testified by the large debris fan in the Rhone valley. Hosting the village of Susten, this partially inhabited debris fan has a radius of nearly 2 km. On yearly average, Illgraben delivers nearly 100,000 m$^3$ of sediments to the Rhone (e.g. Schlunegger

et al., 2009). A large portion of the sediment transfer occurs during debris flow events (Figure 2) making the Illgraben the most active debris flow torrent in Switzerland (Rickenmann et al., 2001).

Debris flows in the Illgraben have been systematically monitored starting in the year 2000. Their observed granulometry and
water content varies between individual events, but they are generally characterized by boulder-rich fronts with limited amount of matrix soil debris and an event main body made up of a finer mixture of liquefied soil debris (Pierson 1986; Badoux et al., 2009; McArdell et al., 2007). The debris flows reach velocities of 4-8 m/s in the lower channel portions and have flow heights of up to 2-3 m (Badoux et al., 2009; Swiss Federal Institute for Forest, Snow and Landscape Research WSL, unpublished data). Flow volumes may range from order $10^3$ to $10^5$ m$^3$. "Small" events not exceeding a few tens of 10
m$^3$ are most frequent and occur up to eight times per year (Hürlimann et al., 2003). Volumes between 75,000 m$^3$ and 250,000 m$^3$ are classified as "intermediate size". Such events occur several times per century and may locally overtop the channel banks. Events classified as "large" can potentially reach populated areas outside the Illgraben channel where they have a particularly high damage potential. This occurred in 1961 when the largest documented flow of ~500,000 m$^3$ destroyed a road bridge on the fan (Badoux et al., 2009). Although no significant channel overtopping in populated areas has occurred
since at least 2000, even smaller debris flow events constitute a threat to lives of people crossing the channel during professional or recreational activities.

Illgraben debris flows have been observed to initiate in the sub-catchment area in the southwest of the catchment (outlined in Figure 1; Berger et al., 2011a), which exposes Triassic schists and dolobreccias as well as quartzites (Schlunegger et al.,
2009). There, erosion on the steep lateral slopes (on average 40°) mobilizes sediments that are subsequently delivered to the Illgraben channel, which are then mobilized to debris flows during intense thunderstorms typically occurring from April to October. The largest debris flow events are expected when temporary creek dams produced by landslides from the steep lateral slopes suddenly fail (Badoux et al., 2009). Much of the debris flow initiation and propagation effects are not fully understood, because the debris flows interact with their surroundings by eroding the Illgraben channel bed (Berger et al.,
2011b) and the channel banks, which in turn may recharge ongoing or trigger additional debris flows (Burtin et al., 2014).

## 2.1 Existing Warning System at the Illgraben

A series of 30 check dams (henceforth, individual check dams are referred to by the letters "CD" followed by a unique number, which increases in flow direction) has been installed along the lower 3.4 km of the Illgraben channel to stabilize the
channel along the current flow path and to minimize channel-bed and lateral erosion (Figures 1 and 2). Instrumentation consists of two separate systems, one for data collection and an independent early warning system for the community. The observation station (Rickenmann et al., 2001, Hürlimann et al., 2003, McArdell et al., 2007) consists of geophones installed on check dams to detect time of passage, flow stage sensors (radar, laser, ultrasonic) to estimate the height of the flow, video

cameras, a vertical wall instrumented with 18 geophone plates (not used in this present study), and a large force plate situated under the roadway bridge near the mouth of the channel (Berger et al., 2011b, McArdell et al., 2007). The observation station is triggered by geophone detection of debris-flow passage at a check dam located approx. 1 km upstream of the force plate and instrumented wall. The geophones measure the vertical velocity of the debris-flow-induced vibrations on the steel plate behind which they are mounted. The signal is logged as impulses, defined as the number of times per second that the geophone signal exceeds a small positive threshold voltage of 0.2 V (e.g. Rickenmann and McArdell, 2007; McArdell et al., 2007; Arattano et al., 2016). The 8 $m^2$ force plate (McArdell et al., 2007) is currently configured to measure vertical and shear forces at a rate of 2 kH (McArdell, 2016). The force plate rests on elastomer elements, which act to partially acoustically isolate the force plate from vibrations in the channel. Apart from the force plate and instrumented wall, batteries and solar panels power all instruments at the observation station.

The existing early warning system at the Illgraben was designed based on experience from the observation station. It has subsequently been optimized to provide reliable early warning for the community (Badoux et al., 2009). The early warning system consists of three rain gauges within and surrounding the catchment, a geophone at the uppermost position in the catchment where instruments are expected to withstand rockfall activity, CD1 (Figure 1), and two geophones and two radar stage sensors at CD9 and 10. Batteries and solar panels power the detection instruments. Warning consists of acoustic alarms and flashing lights installed at channel crossings frequented by tourists, and text messages delivered to the authorities.

Currently, early warning is contingent upon initial detection on the geophones at CD1, 9 and 10. Ideally, this is the geophone installed on CD1. Unfortunately, this system is prone to power outages due to limited sunlight and a weak GSM network signal. In contrast, detections at CD9 and 10 are deemed reliable and are less susceptible to potential damage by rockfall. CD10 also has a laser stage sensor and issues a warning when a predefined flow height is reached. For this warning, delay time defined as the difference between initial detection and debris flow arrival at CD27, ranges from 0 to half an hour and is thus highly variable (Badoux et al., 2009). Finally, flow velocities estimated from propagation between CD10 and 29 typically lie between 1 and 8 m/s.

## 2.2 19 July 2011 Event

In the following analysis we focus on the seismic records of a debris flow event on 19 July 2011. Following the measurement-based method of Schlunegger et al. (2009), we calculated a maximum flow depth of 2.1 m, a flow velocity of about 2.4 m/s and a maximum front discharge of 38 $m/s^2$. Furthermore, with a total volume of around 15,000 $m^3$ this event is classified as small. Initial geophone detection at CD1 occurred on 17:40:08, subsequent detection times are listed in Table 1. After the front passage, the event was characterized by pulse-like flow with around two dozens secondary surges (or "roll waves") arriving over the course of the 15-minute-long event (Figure 3). The individual waves were up to 1 m high, but their

height was variable and diminishes towards the end of the debris flow. The high percentage of fine material encountered in Illgraben debris flows is believed to be responsible for these roll waves (Rickenmann et al., 2001). The flat signal prior to the increase in flow depth (Figure 3) indicates that the voltage is below the threshold value.

## 3 Seismic Data

During summer 2011, a temporary seismometer network (Fig. 1) was operational for about one hundred days (Burtin et al., 2014). The network recorded the seismic signature of several debris flows, including the event on 19 July 2011, on which this study focuses. The seismometers (labeled IGB1-IGB7, IGB9-IGB10; IGB8 was not fully functional on 19 July 2011) were powered by battery and solar energy (sensor and recording specifications are given in Table 2). Ground motion was sampled at 125 or 200 Hz and stored locally. The analysis presented here relies primarily on signal frequencies within the sensors' flat spectral response and for this reason digital counts are converted to ground motion with a single multiplication factor. Burtin et al. (2014) give more details on the seismic instrumentation.

Figure 4 shows the seismic signature of the 19 July 2011 debris flow as well as two additional events in summer 2011 recorded at station IGB02. Burtin et al. (2014) analyzed the event on 13 July 2011; during the event on 29 June 2011 only a few stations of the seismic network were fully operational. All seismograms shown in Figure 4 have typical emergent onsets and slowly fading terminations. The seismograms of the 29 June 2011 event and the 19 July 2011 event analyzed in the present study have comparable vertical ground motion amplitudes and durations of around 30 minutes. For both these events, the in situ measurements provided estimates for debris flow volume and front velocity. Evading in situ detection, the event on 13 July 2011 was likely much smaller, which also explains the weaker seismic ground motion. Its seismogram is somewhat longer and consists of several individual pulses (see Burtin et al., 2014, for details). At IGB01, located near a catchment region where debris flows are believed to initiate (McArdell et al., 2007; Berger et al., 2011a), the 19 July 2011 debris flow signal emerged above the seismic noise at around 17:35 (Figure 5a).

Although the 19 July 2011 debris flow was rather small, it left a strong seismic footprint on all seismometers and occupies a broad seismic frequency range from below 1 to over 50 Hz (Figure 5c). After an additional 10-15 minutes, IGB01 recorded a second event. However, since this event cannot be identified on the other stations, it is likely a local process, such as a landslide near IGB01 or a secondary debris flow, which did not propagate far enough downstream to be recorded at other stations.

Over the course of its duration, the debris flow signal undergoes amplitude variations for two reasons: first, a varying degree of seismic energy generation related to flow velocity, channel topography and granulometry of entrained material and,

second, the changing distance between moving material and recording seismometer. In theory, inter-particle collisions, particle impacts with the channel bed and turbulence in the water-sediment mixture emit seismic waves at all positions along a channel (Tsai et al., 2012; Gimbert et al., 2014). However, the primary seismic source is associated with the debris flow front (Burtin et al., 2014), where large boulders are mobilized (McArdell et al., 2007). This is in agreement with independent studies of bedload transport suggesting that such large grain sizes dominate the energy transmission to the ground, even though their contribution to the overall mobilized volume is small (Turowski et al., 2015). The seismic signal strength can thus be used to trace the debris flow propagation through the seismometer network.

The slowly emerging and fading seismogram envelopes (Figures 4 and 5) yield a typical "tremor-like" appearance, in contrast to impulsive signals associated with, e.g., earthquakes or explosions. Individual signal spikes likely represent impacts of large individual rocks or lateral landslides induced by the debris flow event (Burtin et al., 2014). The emergent character of the 19 July 2011 event is also highlighted in Figure 6, which compares a 2-minute pre-event time series (Panel b) with parts of the debris flow signal of the same length (Panels c and d). During such short time windows, neither amplitude modulation nor arrivals of individual seismic phases are visible making it difficult to distinguish a debris flow record from seismic background noise. However, the relative amplitudes between stations show clear differences: In the pre-event noise record (Panel a), the Rhone Valley stations (IGB3, IGB9 and IGB10) have the largest amplitudes, most likely a consequence of anthropogenic noise (note normalized traces in Panel a). In contrast, near the beginning of the debris flow record (Panel b), ground vibrations are largest in the Illgraben catchment, at stations IGB01, IGB02 and IGB03. Later, when the debris flow has propagated downstream, the valley stations (IGB3, IGB9 and IGB10) record the strongest signal. These temporal and spatial amplitude variations form the basis of the detection and location scheme, which we now describe.

## 4 Detection and Location Scheme

For the 19 July 2011 debris flow, we apply the ASL method (e.g. Battaglia and Aki, 2003; see also Introduction of this manuscript) to locate the source of tremor-like seismic signals via differences in amplitudes throughout the recording array. The amplitude $A_i$ of a seismic signal recorded at the $i^{th}$ station is subject to the decay relationship

$$A_i(r) = \frac{A_0}{r_i^n} e^{-\alpha r_i} \qquad (1)$$

where r is the source-station distance, $A_0$ is the signal amplitude at the source (henceforth "source strength"), $\alpha$ is the signal decay constant and n=1 for body waves and n=1/2 for surface waves (Battaglia and Aki, 2003). Equation (1) describes amplitude decay in the far field, whereas a rigorous representation of source strength naturally has to take into account the

near field. Consequently, $A_0$ may be interpreted as parameterized source strength but lacks a strict physical meaning. In fact, directly at the source location (at r=0), $A_i$ becomes infinite and $A_0$ is undefined.

In Equation 1, the exponential term accounts for anelastic damping of the seismic wave, whereas the $1/r^n$ factor describes
5   amplitude attenuation due to geometric spreading. The decay constant $\alpha$ can be expressed as

$$\alpha = \frac{\pi f}{Q\beta}$$ (2)

where f is the signal frequency, Q is the seismic quality factor and $\beta$ the seismic wave velocity.

The essence of the ALS technique is to measure the amplitude $A_i$ of a seismic signal on several seismometers. Ideally, the seismometers locate at different distances to the signal's source to yield a large spread in measured amplitudes. Subsequently, Equation (1) is used to model the different recorded amplitudes throughout the network. As the source location, attenuation $\alpha$ and source strength $A_0$ are unknowns, these quantities have to be determined via inversion of
Equation (1). Consequently, when grid searching over potential source locations, the grid point corresponding to the minimum misfit of Equation (1) to the measured amplitudes indicates the source location.

Rather than using instantaneously recorded amplitudes of the seismic ground vibration, we calculate the signal's root mean square (RMS) amplitudes at each recording station for a specified time window. The RMS is a time-averaged strength
measure of the debris flow signal and a robust measure of induced ground motion whose spatial variations throughout the array are subject to Equation 1. It should be stressed that in volcanic applications, site amplification (or damping) has demanded seismic signal correction prior to application of Equation 1 (Aki and Ferrazzini, 2000; Battaglia and Aki, 2003). For our Illgraben data, site amplification effects on the ASL performance seem minor and are discussed below.

We apply Equation (1) for the case of body waves, which is in agreement with the geometric spreading corrections applied by Burtin et al. (2014) for the same seismic network. Topography and vertical seismic velocity gradients below the surface result in curved ray paths for both surface and body waves, respectively. However, as we cannot constrain the catchment's seismic velocity model, we cannot estimate this curvature for body waves and we approximate the ray path between debris flow front and a given seismometer by a straight line. Using the straight-line approximation for surface waves, Equation (1)
also produced reasonable locations of the debris flow front. We did not investigate the advantages and disadvantages of using body waves instead of surface waves systematically. Such a comparison is planned in a future study when more debris flow records and possibly active source seismograms for ground truth comparison are available.

For our decay-fit locations, the geographic location and source strength of the debris flow front is varied in a grid search. Moreover, we vary alpha between 0 and 0.001, which corresponds to no attenuation and a high quality factor of 200 for S-waves at velocities of 863.94 m/s. A dominance of S-waves over P-waves is reasonable, because we are confining the analysis to vertical seismogram traces. For shallow seismic sources, such as debris flows, P-wave particle motion should be strongest on the horizontal components and weaker compared to vertically polarized S-waves. Vertically polarized P-waves will become stronger as ray path curvatures increase for strong vertical seismic velocity gradients as well as larger source-station distances and elevation differences, but this effect is assumed to be of secondary importance. Ground motion amplitude is estimated via the root-mean-square of 100-second seismogram time windows and $A_0$ is varied between 500 and 1500 times the root-mean-square maximum measured throughout the network. Fit quality is quantified with the variance reduction defined as

$$VR = \left(1 - \frac{\sum(data - fit)^2}{\sum(data^2)}\right) * 100\%, \qquad (3)$$

where in our case "data" refers to the root-mean-square of the 100-second seismogram time windows and "fit" refers to the calculated ground motion (Equation 1). The summation is carried out over the nine available seismometers. Equations 1 and 3 are applied in a spatial grid search over geographic coordinates and the maximum in variance reduction (100% represents a perfect fit) indicates the source of the recorded seismic signal. We refrain from constraining the search grid to the torrent channel and instead determine the location of the debris flow front by projecting the seismic source to the channel coordinates.

The VR absolute upper limit (100%) facilitates interpretation of fit quality and its variation at different times. The disadvantage of using variance reduction as an indicator for fit quality is that for a monotonic fitting function such as Equation (1), even poor fits may provide relatively high variance reductions (~80% as shown below).

**5 Results: Seismic Noise Sources and Debris Flow Locations**

Prior to decay fit location, we apply a two-pole 0.5-5 Hz Butterworth bandpass filter to the seismic time series, acknowledging that the debris flow seismograms primarily exhibit higher frequencies (Figure 5). However, the chosen frequency range is a compromise between minimizing effects of spatial differences in decay constant α and staying near a range where the debris flow transmits seismic energy and the frequency response of our sensors is flat (Table 2).

The results of fitting decay curves (Equation 1) to consecutive 100 s amplitude averages on 19 July 2011 (including the debris     flow     event)     are     illustrated     in     the     animated     movies     in     the     supplemental     material

(http://people.ee.ethz.ch/~fwalter/download/movies/df/), in Figure 7 and in Figure 8. Between 01:00 and 04:00, the variance reduction lies between 80 % and 90 % and calculated seismic source strength is low. Consequently, even in the absence of a dominant seismic source, variance reductions above 80 % can be expected. At around 04:00, the variance reduction rises and approaches 100% and the source strength increases as well, though by less than an order of magnitude. This marks the influence of a noise source, whose signal is detectable on station IGB07 in the upper Illgraben catchment as well as station IGB09 on the debris fan (Figures 7c and 8c and 9a). Decay-curve fitting locates the source of this persistent noise signal between IGB03 and IGB10 within or near the village of Susten, suggesting an anthropogenic noise source. Despite temporary drops in variance reduction to 80 % or lower accompanied by drops in source strength (Figure 9), this source continues dominating the noise field throughout the afternoon.

After 15:00, the noise source strength fades and fluctuating variance reductions indicate that there no longer exists a single noise source dominating the entire array. Near 17:35, the variance reduction and the source strength increase, the latter drastically by almost two orders of magnitudes. This marks the beginning of the debris flow event. The signal source locates high up in the catchment area of the Illgraben torrent (Figure 7 and movies in the supplemental material). During most of the following 100-second time windows, the decay fit determines locations with variance reductions near 100%.

During the debris flow event, the decay-fit locations projected onto the channel move downstream at an average of 1.8 m/s (movies in the supplemental material and Figure 10). For comparison, the geophone-derived arrival times at CD1, 10, 24 and 27 yield an average velocity of 2.9 m/s (Fig. 10). Furthermore, whereas the CD1 arrival time for the geophone detection and the seismic decay-fit location nearly coincide, the arrival time differences grow as the debris flow moves downstream. At CD24 the decay-fit arrival time lags 10 minutes behind geophone detection. We interpret this discrepancy to result from changes in seismogenic processes within the debris flow: near the initiation, the debris flow front primarily transmits the seismic energy. At each of the 100 s time windows, the ASL scheme locates this moving "point source". Subsequently, the debris flow spreads out longitudinally and later arriving debris flow parts participate in the seismic transmission. Rather than a point source, the ASL-derived location may now locate a volumetric centroid of the seismogenic debris flow part. As a result, the later arriving debris flow parts bias the decay fit locations backward from the debris flow front. This interpretation is supported by later arriving roll waves (Figure 3), each transmitting seismic energy, as well as typical changes in longitudinal debris flow profiles, which become progressively less steep and thus stretched out during propagation (Berger et al., 2011b). The influence of heterogeneous subsurface geology beneath the upper catchment and debris flow fan seem to have a minor influence as argued below.

The distance from the variance reduction maximum to the Illgraben channel varies between below 100 m and nearly 900 m (Figure 10b). Assuming that the debris flow source is confined to the channel, these numbers provide an approximate measure for location uncertainty. During the debris flow, the decay constant $\alpha$ takes values within the entire grid search

range (0 - 0.001 m$^{-1}$). In view of the local topography and ground structure heterogeneities within the grid search area, it is difficult to interpret the spatially averaged value for α and its variations. However, they likely do carry physical meaning, because when fixing α=0, variance reductions drop by 10-20% during several 100 s time windows.

5    During the first 5 minutes of the debris flow, the seismic source strength grows slowly (Figure 10a). In the 100 s time windows starting between 17:31 and 17:32, the source strength reaches $1.8 \times 10^{-4}$ m/s and thus exceeds the strength of other seismic sources in the upper catchment measured earlier that day (Figure 10b). By 17:40:01, near the time of geophone detection at CD1 (17:40:08; Table 1), seismic source strength of the debris flow has increased by around an order of magnitude.

Whereas during the time window starting at 17:32 the decay fit unmistakably locates the debris flow source to the upper Illgraben catchment, higher values of source strength do exist in prior time windows. Nevertheless, on the day of the debris flow, these higher values do not correspond to high quality locations (high variance reduction) in the upper catchment. For the time window starting at 17:40, the decay fit locates the debris flow source at CD1, which is confirmed by the independent geophone detection (Figure 10b). Given these observations, we interpret the time window starting at 17:32 as the earliest seismic detection of the debris flow with our decay-fit approach. The processing of the entire debris flow day already indicates that source strength, source location and decay fit quality (variance reduction) all have to be considered simultaneously in order to reduce false detections. In the following, we investigate this systematically for a ten-day period.

20   **6. Debris Flow Detection: Robustness and Potential Improvements**

Source strength, variance reduction and location calculated with the ASL method should be combined in debris flow detection schemes. When used separately, these parameters are not robust enough. In particular, variance reduction can be misleading due to the curvature of the decay function (Equation 1). At large distances, all amplitude measurements fall into the nearly flat part of the distance-decay curve and in this case, a set of similar amplitude measurements from different stations will yield a high variance reduction. Ideally, a network should be designed such that amplitude measurements of debris flows cover distances, where the decay relationship has a range of slope values (e.g. Figure 7). Such cases can be easily distinguished from signals of far-away sources, whose decay throughout the network is negligible, because nearly parallel wave fronts have a negligible geometric spreading attenuation ($1/r_i^n$-term in Equation 1).

30   In order to test the robustness of the ASL-based detection of the 19 July 2011 debris flow, we calculate the debris flow locations with individual stations removed and process a ten-day period between 14 and 23 July 2011. During this period, all stations used in the above analysis were operational, except for a one-hour window (00:00-01:00 UTC on 19 July 2011). We

define potential debris flow detection as those 100 s time windows, whose ASL fits give a variance reduction of 90 % or higher and a source strength $A_0$ exceeding 1.7e-4 m/2. Moreover, locations have to fall within the upper catchment (cyan dashed lines in Figures 7 and 8). These conditions were chosen such that the debris flow record satisfies them at 17:32 on 19 July 2011, which above we defined as the earliest ASL-based detection.

## 6.1 Station Removal

We test the robustness of the calculated source strengths $A_0$ against the removal of individual upper catchment stations (Figure 11). $A_0$ is most sensitive to the removal of station IGB02, which lies closer to the torrent than any other upper catchment station (IGB01-IGB07). For some times during the debris flow, removal of station IGB02 leads to a drop of $A_0$ by nearly an order of magnitude. Removing station IGB01 tends to increase $A_0$, although by a smaller amount. In contrast, removing other upper catchment stations has minor effects, which often fall within the resolution of the grid search inversion for $A_0$. The sensitivity of calculated $A_0$ values to removal of IGB01 and IGB02 increases as the locations of the debris flow front move downstream and thus likely has to do with the proximity of these stations to the passing debris flow front. Near the event beginning, when the debris flow is not particularly close to IGB01 or IGB02, the $A_0$ values are more stable, and removing single stations can lead to a 1-2 minute delay in detection.

## 6.2 False Detections

During the ten-day period (14 to 23 July 2011) we obtained altogether 39 false detections. 25 of these locate conspicuously close to a seismometer, most of them close to IGB07, where either electronic spiking or local seismic noise often causes high-amplitude signals. Two false detections are associated with earthquake-like signals. Six false detections had locations north of IGB02 so they do not strictly locate in the upper catchment. Of the remaining false detections, two located close to IGB06. As visible inspection confirms good decay fit quality, these two detections may be associated with local landslide or rock fall activity. Finally, there exist four remaining detections, which we call "unclassified".

Anticipating a strong influence of extreme amplitudes at a single station, we reprocess the 25 detections whose locations are close to a particular seismometer and the four "unclassified" detections after removing the closest station to the calculated location. Of the 25 false detections, 23 move down valley and thus outside the region we define as upper catchment. The other two still end up in the upper catchment, but with substantially lower variance reduction and/or source strength. For the four "unclassified" detections, either the epicentres shifted after station removal or visual inspection showed a low decay fit quality despite a high enough variance reduction to initially trigger detection.

We conclude that of 39 false detections, 37 are associated with poor performance of the ASL technique (including detections of two earthquake signals) and only two may be due to geomorphological activity in the catchment. The effort needed to identify false detections seems reasonable. In most cases, testing the effects of single station removal can be used and likely be implemented automatically. For other cases, visual inspection of the amplitude decay fit can quickly provide clues about the meaningfulness of the detection. Furthermore, even teleseismic earthquake waves can trigger ASL detection, though not necessarily in the upper catchment region (Figure 9). Consulting real-time records of permanent earthquake monitoring seismometers can help identify such false detections.

## 6.3 Time Window Length

The choice of time window length naturally affects early warning time, because a warning can only be issued at the end of the time window and after data transmission. We initially chose 100 s long time windows, because this results in a smooth downstream propagation of the calculated location of the debris flow front. Figure 10 also shows the locations for 30 s windows, which still agree reasonably with the 100 s window locations. However, our attempts to use the ASL technique to locate short (order second long) rock fall events documented in Burtin et al. (2014) were unsuccessful. This is somewhat surprising because Kumagai et al. (2009) successfully located debris flows on Cotopaxi Volcano (Equador) using 5-second long time windows. We speculate that the pulse-like nature of our rock fall signals induces ground vibration, which is not well represented by our root mean square metric. In any case, we suggest that there is a minimum window length for the ASL technique, which should be systematically investigated for different debris flow and other mass motion signals.

## 6.4 Site amplification

So far we neglected site effects, which may amplify or diminish seismic amplitudes at individual stations. Coda amplification factors derived from earthquake records can be used to correct for such site amplification, which has been applied to debris flow monitoring using the ASL technique (Kumagai et al., 2009). This method is rooted in the scattering nature of local earthquake coda, which implies that later coda parts depend on geology near a seismometer installation, but are independent of the path between earthquake source and recording seismometer (Aki and Chouet, 1975). Consequently, the ratio of site amplification between two stations can be calculated using envelope ratios of earthquake S-wave coda recorded at the same two stations. To insure full independence from path effects, envelopes of coda waves whose arrival times exceed roughly twice the S-wave travel time are typically used (for review on technical details and theoretical fundamentals see Sato et al., 2012).

We calculate coda amplification factors using local earthquakes whose coda amplitudes are still above the background noise level after two S-arrival times. Unfortunately, only three earthquakes passed this requirement and were recorded with more

than 5 stations (Table 3). We do not consider teleseismic signals whose earthquake sources locate at larger distances, because secondary arrivals may mix with the S-wave coda.

Coda envelopes calculated by root-mean squares of 2.5 second-long windows of the best-recorded earthquake (M2.2 at 12 km distance) are shown in Figure 12a. Panel b of the same figure shows two-second averages of all smoothed envelopes normalized against measurements at station IGN04. This station was chosen, because according to visual inspection, its records of all earthquakes are among the cleanest ones. The coda envelope ratios show substantial fluctuations, however they follow qualitative trends: stations IGB02 and IGB03 show the strongest amplification, and the Rhone valley stations IGB09 and IGB10 tend to have the weakest amplification. To explain these differences in amplification requires analysis of heterogeneous subsurface geology in combination with topography. This is beyond the scope of this paper, but the stations' subsurface geological structure is expected to have a larger effect than topographical characteristics (Burjánek et al., 2014).

Our range of amplification values is rather narrow compared to Kumagai et al. (2009), who found values between 0.4 and 1.8. Moreover, our amplification factor uncertainties clearly exceed their uncertainties, for some stations by more than an order of magnitude. This may be the result of the relatively weak earthquakes that were available for our study. We nevertheless apply the site amplification correction based on the results shown in Figure 12. With respect to IGB04, we group our network into three classes: IGB02 and IGB03 amplified by a factor of 1.5 and IGB01, IGB09 and IGB10 amplified by a factor of 0.75. The amplitudes of the remaining stations are not changed. Given the large uncertainties in coda amplification and the limited available earthquake records, we feel that this first order correction is most reasonable.

The effect of site amplification correction on the ASL performance is minor (dashed lines in Figure 10). Compared to the initial calculations, the locations of the debris flow front propagate at similar velocities and distances from the torrent bed. Only for some times after 17:50, propagation is somewhat faster and thus better matches the in situ detections (Figure 10b). This suggests that the delayed debris flow arrival times calculated on the fan are mostly a result of another effect, such as a longitudinal stretching of the debris flow profile as suggested above.

## 6.5 Background Noise

Because the present detection scheme relies entirely on amplitude information and neglects signal phase, its success is particularly dependent on levels of seismic background noise. Only stations where the debris flow signal emerges above the background noise level are of use to the decay fit.

We evaluate how changes in seismic background noise may affect debris flow detectability by comparing seismic signal strengths recorded at IGB09 and IGB10 to a noise record of the additional seismometer station IGN01 (Figure 1). This

station (Type Lennartz LE 3D 5s sensor; flat frequency response: 0.2-50 Hz; sampling frequency: 200 Hz) was operational between 27 May 2015 and 16 July 2015 and was installed in the Rhone Valley, some 500 m west of the Illgraben channel. The original purpose of this station was to record a debris flow seismogram in a quieter location than stations IGB09 and IGB10, which, according to our decay fit locations, were installed closer to dominant anthropogenic noise sources. Unfortunately, no debris flow occurred during the deployment of station IGN01. Nevertheless, this station's record is well suited for a comparison with background noise at the other Rhone Valley stations IGB09 and IGB10.

To characterize the background noise floor and its variations, we followed the procedure of McNamara and Buland (2004). We divided continuous records of IGB09, IGB10 and IGN01 into 10-minute long windows and for each window we calculate the power spectral density (PSD) from the Discrete Fourier Transform. PSD was calculated in units of decibel with a reference ground velocity of 1 m/s. The hourly averages of the 10-minute PSD's are subsequently distributed between -100 and 200 dB into 0.5 dB-wide bins from which probability density functions (PDF's) of PSD are calculated.

Figure 13a shows the 51-day-long noise PSD-PDF recorded at IGN01 and the mean and standard deviation of a 19-hour-long noise PSD-PDF recorded at IGB10, which includes the debris flow. At both stations, the noise level is comparable, while peak probabilities in the IGN01 PSD-PDF lie below one standard deviation of the IGB10 noise mean. This supports the expectation that during substantial time periods, IGN01 is quieter than IGB10. Figure 13b shows again the 51-day-long noise PSD-PDF of IGN01 together with the debris flow spectra recorded at stations IGB09 and IGB10. Within the 1-5 Hz frequency range relevant for our decay fit locations, the PSD-PDFs show two branches in noise amplitude (marked with two arrows). The branches are separated by up to 12 dB and reconnect above 5 Hz. Extracting PSD curves, which are bundled in the upper PDF branch (not shown) associates this stronger branch with typical working hours during the week and thus the main contribution of anthropogenic noise. However, the 19 June 2011 debris flow signal recorded at IGB09 and IGB10 dominates this anthropogenic noise more than 90% of the 51-day deployment period of IGN01 (Figure 13c).

This noise analysis uses a single spectral representation of the debris flow seismogram averaged over the entire event duration. Nonetheless, it does indicate that even in the Rhone valley, where anthropogenic sources prevail, the 19 July 2011 debris flow signals dominate the seismic spectrum compared to continuous records of seismic background noise. As most of our seismic monitoring network was located further away from the strong anthropogenic sources, excluding stations in the presence of dominant anthropogenic noise does not seem necessary. Therefore, we suggest that for debris flow events, whose seismic source strengths are at least as high as the 19 July 2011 event, anthropogenic noise does not affect detectability using the ASL method proposed here.

In this detectability analysis we assumed that background noise did not change significantly between deployment periods of stations IGB09 and IGB10 (2011) and station IGN01 (2015). New or temporary construction sites, differences in traffic flow

or other factors would violate this assumption and argue once more for valley seismometer installations away from the village of Susten or the main highway parallel to the Rhone River (Figure 1). Ideally, a noise analysis should be constantly updated and repeated throughout a monitoring period.

**7 Discussion: Suitability for Early Warning**

Applied to debris flows, the ASL method requires no user interaction (such as seismic phase arrival identification) or seismic velocity model and in the Illgraben case performs reasonably even without site amplification correction. As previous authors have suggested, these features recommend the method for automation and thus as a potential ingredient in early warning systems (Kumagai et al. 2009; Ogiso and Yomogida 2015; Jolly et al., 2002). Another strength of the ASL method lies in its ability to detect and locate debris flows in the upper catchment, where in-torrent or near-torrent instrumentation is not

feasible.

The present analysis focused on a single debris flow and more records are necessary for a rigorous performance evaluation. However, we can already make some statements about the strengths and weaknesses of the ASL method and how it can improve debris flow warning even in well-instrumented catchments, such as the Illgraben. The large number of false

detections (39 in ten days) demands post processing of ASL results and most of this can be automated via testing the effects of single station removal. Most remaining false detections can be straightforwardly identified if a person on duty visually checks the decay fit quality and eliminates the possibility of earthquake triggering by comparing seismograms of catchment stations to permanent online seismometer installations. With more sophisticated algorithms and fit quality quantifiers, this could also be automated. Detection reliability could furthermore be improved with help of infrasound arrays whose

automatic performance for Alpine mass motion detection tends to be robust (e.g. Preiswerk et al., 2016).

Given our ASL calculations, we concluded that the first possible debris flow detection occurred during the 100 s time window starting at 17:32. For automated alarms, using smaller time windows will improve early warning times. Testing 30 s-long windows gave promising results (Figure 10). For comparison, data transmission and processing requires less time: For

instance, 90 % of the data streams of the Swiss Seismological Service are transmitted as 1-2 s-long packages within 6 s or less (R. Racine, personal communication). Our grid search is currently implemented in Matlab® and runs on a single processor. It takes less than 3 s to process a 100 s time window. This time could be further reduced by distributing computation on several processors and/or by limiting the grid search to the vicinity of the torrent channel. Similarly, a search domain, which avoids locations, where seismometers reside in the near field would reduce the grid space in addition to

providing numerical stability. Essentially, the ASL locations of the debris flow front seem reliable in the upper catchment, although they lag behind the in-torrent detections by as much as 10 minutes on the lower reaches of the debris fan. Focusing on the early portions of the debris flow seismogram, ASL detection at Illgraben can thus likely improve early warning time

with respect to the in-torrent sensors by several minutes. Considering that the uppermost in-torrent instrumentation is subject to frequent malfunctioning, this could be a decisive advantage.

Finally, it should be pointed out that the ASL method likely performs differently for other types of debris flows. In contrast to our 19 July 2011 event, the weaker 13 July 2011 event exhibits individual flow pulses (Figure 4) and several brief (second-long) rock-fall signals, as documented by Burtin et al. (2014). This may be the reason why our attempts to detect, locate and trace the front of the 13 July 2011 debris flow were less successful. In this context, it is interesting to note that during the two events precipitation was comparable (13 July 2011: 32.4 mm of total rainfall with a maximum rainfall intensity of 3.0 mm/10 minutes; 19 July 2011 event: 22.6 mm total rainfall with a maximum rainfall intensity of 2.6 mm/10 minutes; Swiss Federal Institute for Forest, Snow and Landscape Research WSL, unpublished data). This implies that similar precipitation events resulted in different kinds of debris flows, namely one weak event exhibiting flow pulses and rock-fall signals (13 July 2011) and one event, which had a stronger seismic signature but lacked the flow pulse and rock-fall signals (19 July 2011). These different debris flow responses could be explained with different triggering mechanisms: Rainfall during the 13 July 2011 event triggered widespread lateral slope failure resulting in numerous landslides and rock falls. This may have been possible, because previous precipitation had increased pore water pressure in the ground to a critical level. Interaction between landslides and the debris flow then produced several flow pulses (Burtin et al., 2014). In contrast, the 19 July 2011 debris flow seismograms show little (if any) landslide signals, which can be explained by less loose material in the lateral slopes or absence of critically elevated pore water pressures, or both. In contrast to the 19 July 2011 event, the 13 July 2011 debris flow may thus be classified as "landslide-triggered". It remains to be shown if the ASL method is in general more successful when applied to debris flows, which are not landslide-triggered.

**8 Conclusion**

The amplitude source location (ASL) method presents a promising approach for automated debris flow detection. Our proposed implementation of the ASL method uses exclusively averaged amplitude information. This provides efficient location and rapid detection of debris flows as soon as their seismicity dominates ground vibrations throughout a catchment-wide seismic network. Technical challenges for data communication and processing remain and our approach would clearly benefit from concurrent monitoring with independent methods. Notwithstanding, the ASL technique successfully detected the initiation of the 19 June 2011 debris flow at Illgraben and traced the propagation of its front towards the valley. The simple and efficient decay-fit processing reduces user interaction, requires no seismic velocity model and gives flexibility for locations of seismometer installation. This makes the ASL approach a promising candidate for operational early warning systems against debris flow hazards.

## Acknowledgements

The salary of FW was partially covered by the SNF grant PP00P2_157551. The SEIS-UK equipment pool (NERC) and the École et Observatoire des Sciences de la Terre de Strasbourg provided the seismic stations deployed during summer 2011. We thank Jérôme Vergne for help with instrumentation and network planning. Seismic equipment and data archiving facilities for station IGN01 were provided by the Swiss Seismological Service. Yvo Weidmann helped with the preparation of the supplemental material. The comments of Andrew Lockhart and two anonymous reviewers greatly improved the quality of our study and manuscript.

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

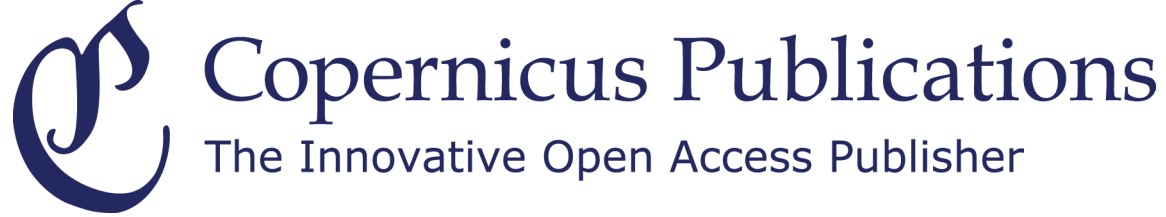

**Figure 1: The logo of Copernicus Publications.**

TABLES

| Check Dam ID | Arrival Time |
|---|---|
| CD1 | 17:40:08 |
| CD10 | 17:43:10 |
| CD24 | 17:55:48 |
| CD27 | 17:58:22 |
| CD29 | 18:02:16 |

Table 1: Arrival times of debris flow front at check dams instrumented with geophones.

| Station Name | Sensor Type | Range of Flat Frequency Response (Hz) | Sampling Frequency (Hz) |
|---|---|---|---|
| IGB01 | Güralp CMG-6TD | 1-100 | 200 |
| IGB02 | Güralp CMG-40T | 0.033-50 | 200 |
| IGB03 | Güralp CMG-6TD | 1-100 | 200 |
| IGB04 | Güralp CMG-6TD | 1-100 | 200 |
| IGB05 | Güralp CMG-6TD | 1-100 | 200 |
| IGB06 | Güralp CMG-6TD | 1-100 | 200 |
| IGB07 | Güralp CMG-6TD | 1-100 | 200 |
| IGB09 | LE-3Dlite | 1-100 | 125 |
| IGB10 | LE-3Dlite | 1-100 | 125 |

Table 2: Specifications for seismic network instrumentation used in detection and location scheme of the present analysis.

| Time | Magnitude | Lat / Lon (° / °) | Distance to Epicenter (km) |
|---|---|---|---|
| **2011-07-15 03:29:03** | 2.2 | 46.22 / 7.74 | 12 |
| **2011-08-21 19:39:45** | 2.9 | 46.04 / 6.89 | 60 |
| **2011-09-06 12:18:57** | 1.6 | 46.28 / 7.24 | 28 |

Table 3: Source parameters of earthquakes used in the site amplification analysis.

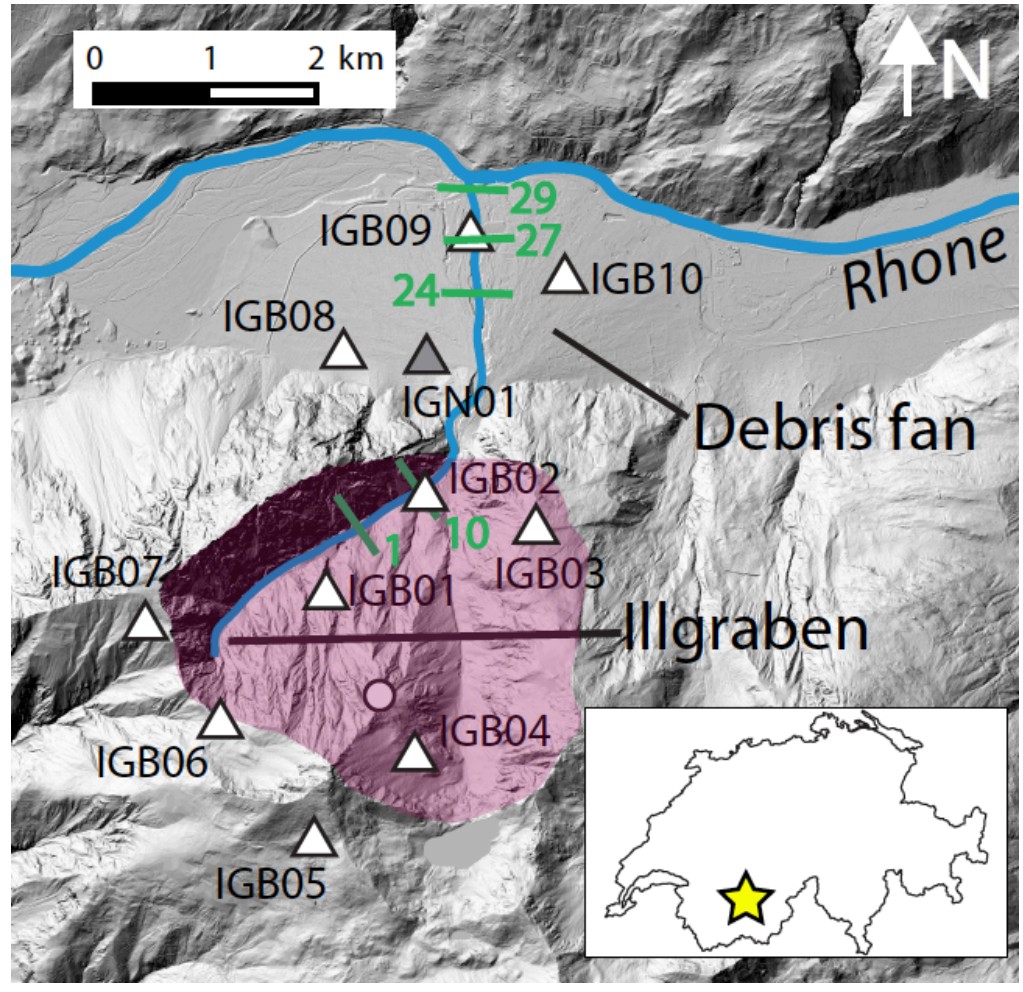

Figure 1: Illgraben region with its upper catchment area (shaded polygon) and debris fan in the Rhone Valley. Numbered green lines represent Check Dams 1, 10, 24, 27 and 29; 2011 seismometer locations are indicated by triangles (grey triangle represents the 2015 noise record station) and a WSL rain gauge is indicated by the circle. Inset shows outlines of Switzerland and yellow star marks location of Illgraben.

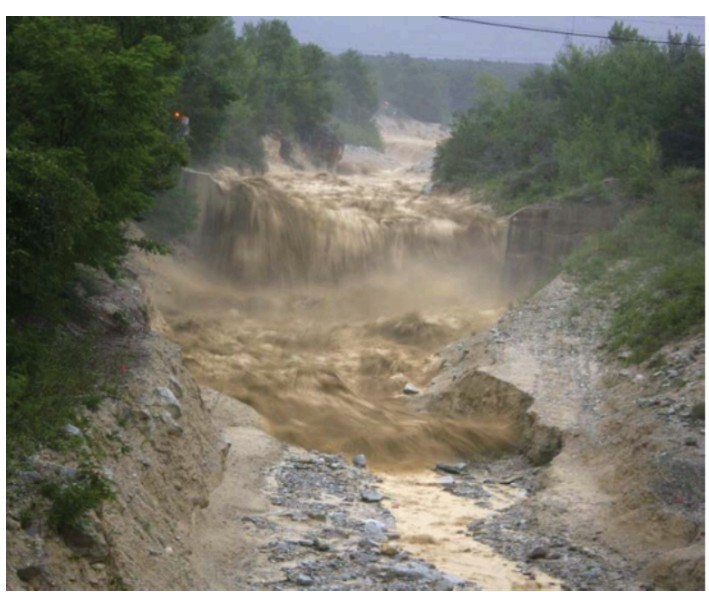

Figure 2: Photo of Illgraben debris flow event near Check Dam 28. Source: Brian McArdell, WSL.

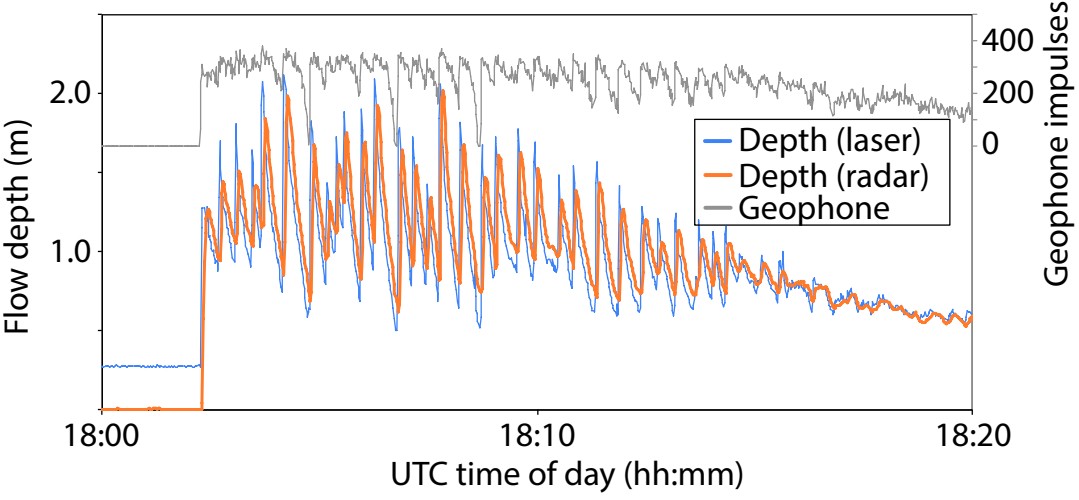

Figure 3: Flow depth and geophone impulses of Illgraben debris flow event on 19 July 2011 (recorded near Check Dam 29).

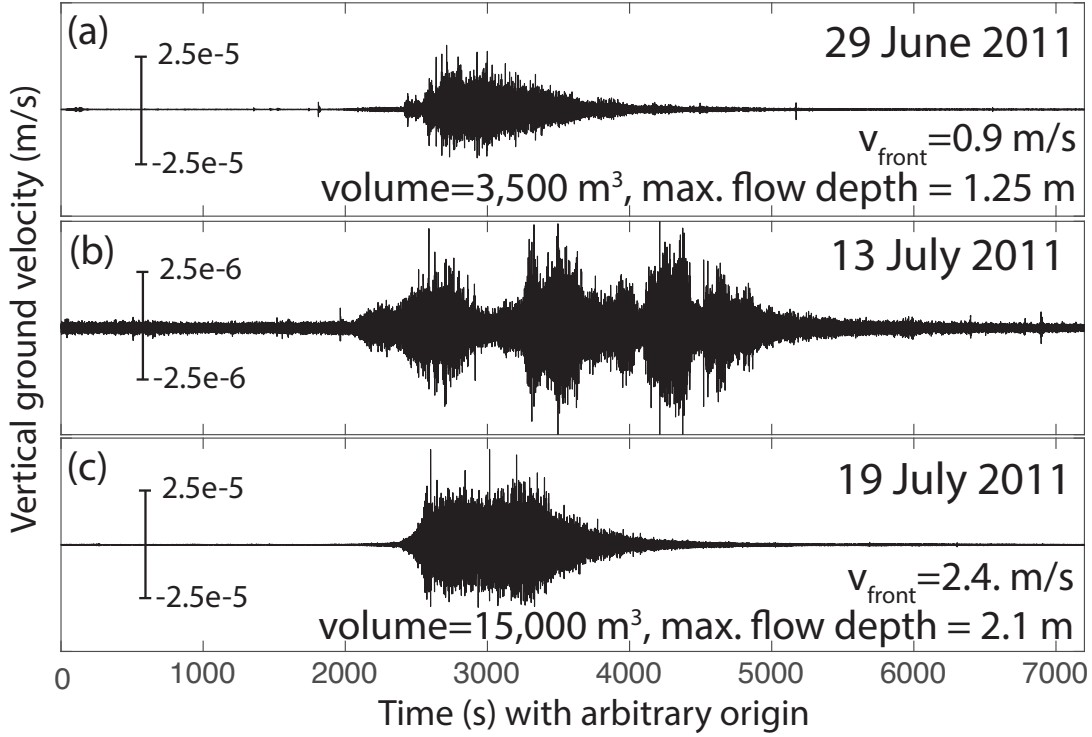

Figure 4: Seismograms of three debris flows recorded at station IGB02. The event on 19 July 2011 (c) is analyzed in the present study. Burtin et al. (2013) focused on the event on 13 July 2011 (b). Note the different y scales. Debris flow parameters in Panels (a) and (c) are calculated following Schlunegger et al. (2009) using unpublished data by the Swiss Federal Institute for Forest, Snow and Landscape Research WSL. All shown time series were filtered between 0.5 and 5 Hz.

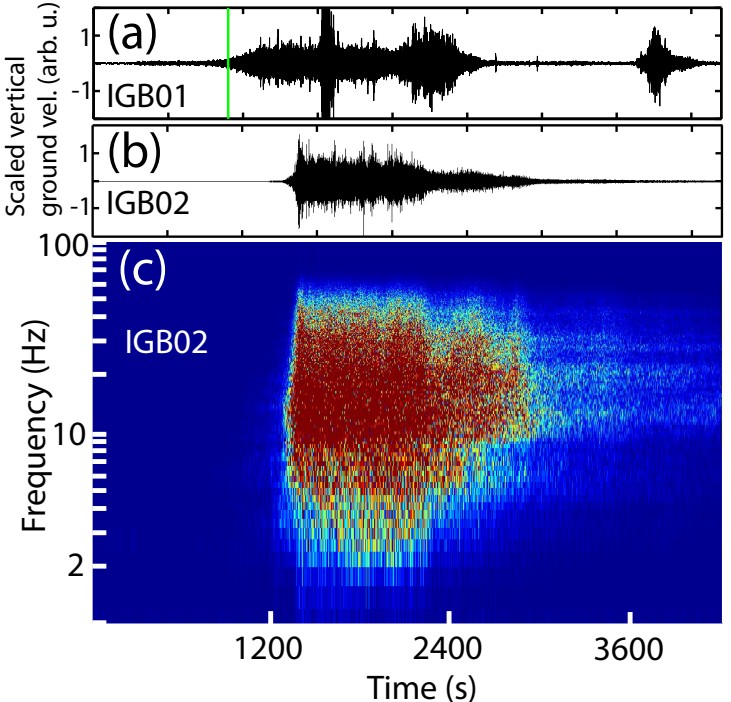

Figure 5: Debris flow seismograms at stations IGB01 and IGB02 (Panels a and b, respectively) and spectrogram of station IGB02 (c). Green bar in Panel a denotes 17:35 on 19 July 2011. Note the second seismic burst after 3600 s likely representing a local mass motion event near IGB01. For illustration purposes, a band pass filter between 0.01 and 50 Hz has been applied to the time series (IGB01 and IGB02 are flat between 1 and 100 Hz and between 0.033 and 50 Hz, respectively, as specified in Table 1).

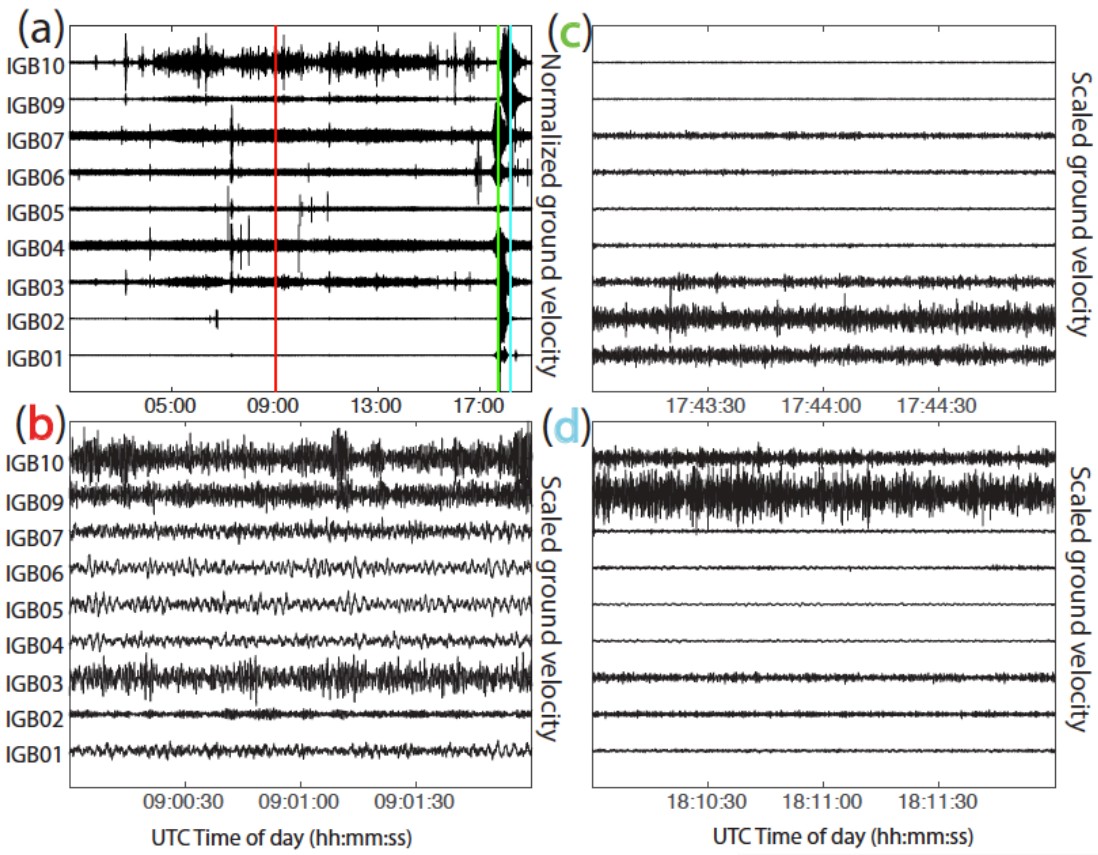

Figure 6: Debris flow seismograms and pre-event noise for the event on 19 June 2011. (a) Record showing the debris flow event around 18:00 and pre-event background noise. (b), (c) and (d) show two-minute long records at the time instances denoted by the red, green and cyan bars in Panel (a). The different amplitude distributions during the two debris flow records (Panels c and d) testify to the motion of the seismogenic debris flow front through the monitoring network. Note that amplitudes in Panel (a) are normalized to each trace, whereas amplitudes in Panels (b), (c) and (d) are normalized to the maximum across all traces.

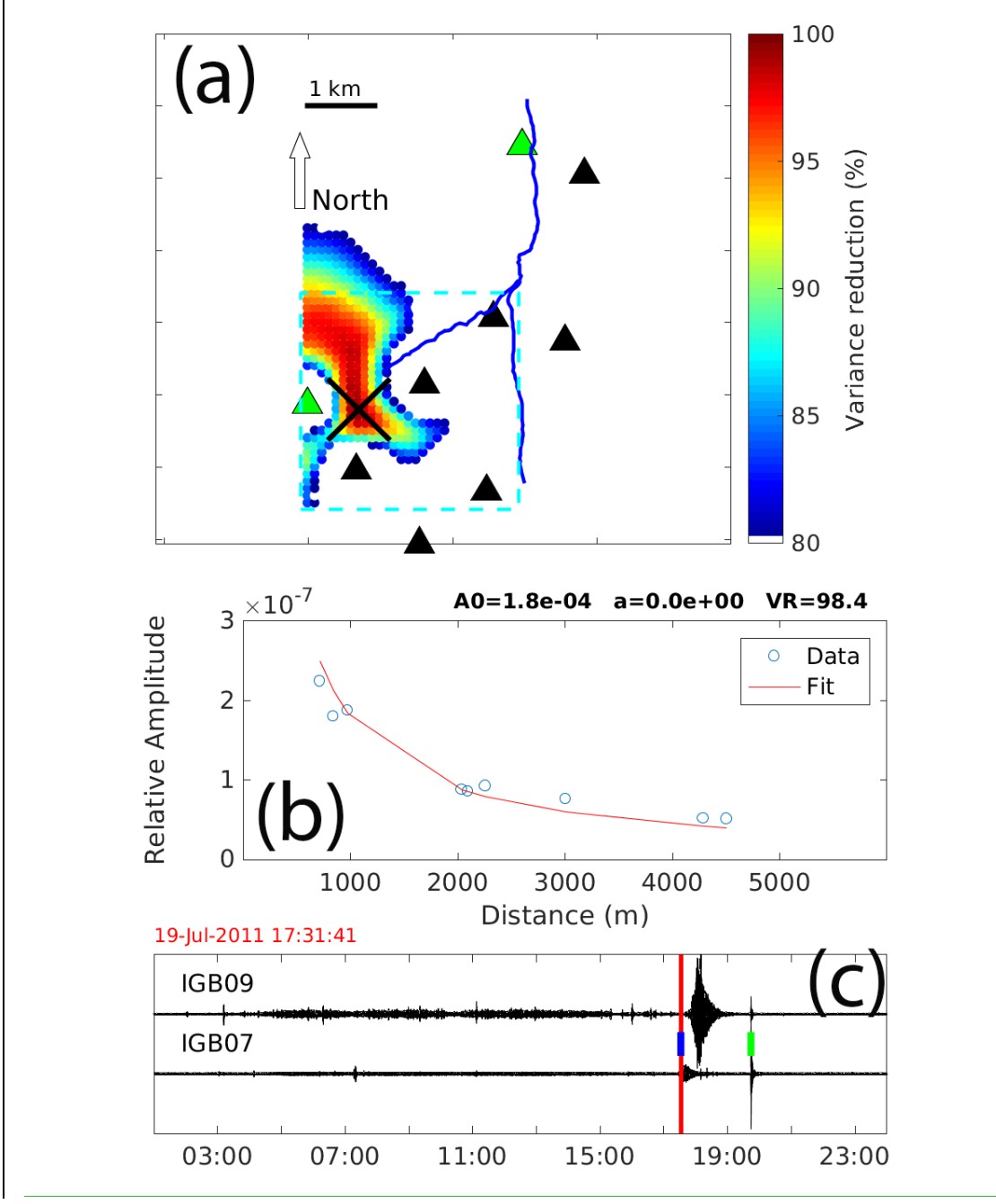

Figure 7: Decay-fit location of debris flow front at the initiation of the event. (a) shows the seismic network, the torrent channel (blue line) and color-coded grid locations for variance reductions exceeding 80%. Black cross indicates best-fit location. The dashed cyan box outlines the region that defines detection in the upper catchment. (b) shows the amplitude attenuation fit associated with the best-fit location. (c) shows the time instance (red bar) on an Illgraben catchment seismometer record (IGB07) and a Rhone Valley seismometer record (IGB09), both marked in Panel a with green triangles (IGB07: southern station, IGB09: northern station). Small blue and green bars denote the beginning of debris flow event and arrival of teleseismic signals from the M6.2 2011 Fergana Valley earthquake in Kyrgyzstan.

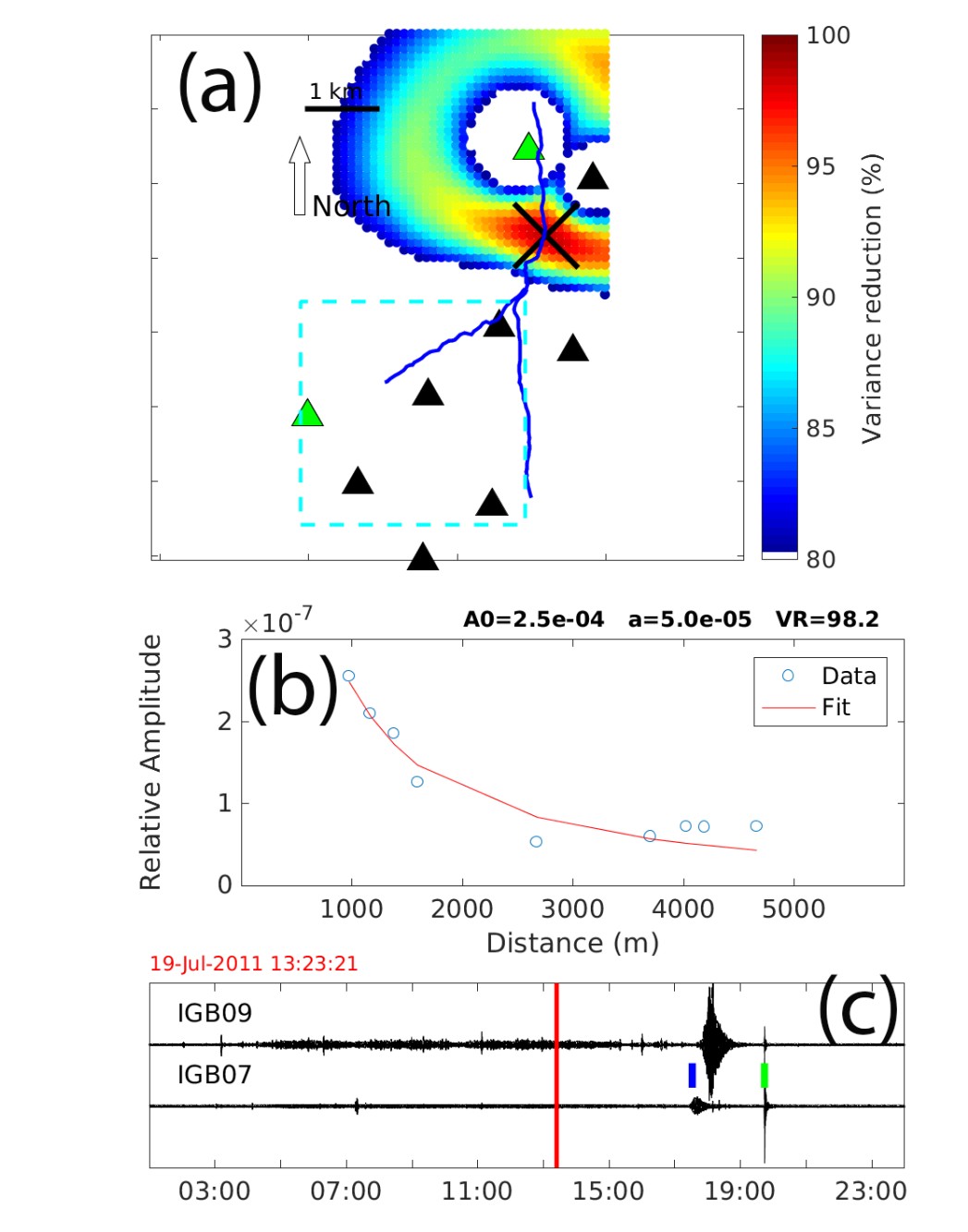

Figure 8: Same as Figure 7, except during a noise window before the debris flow event.

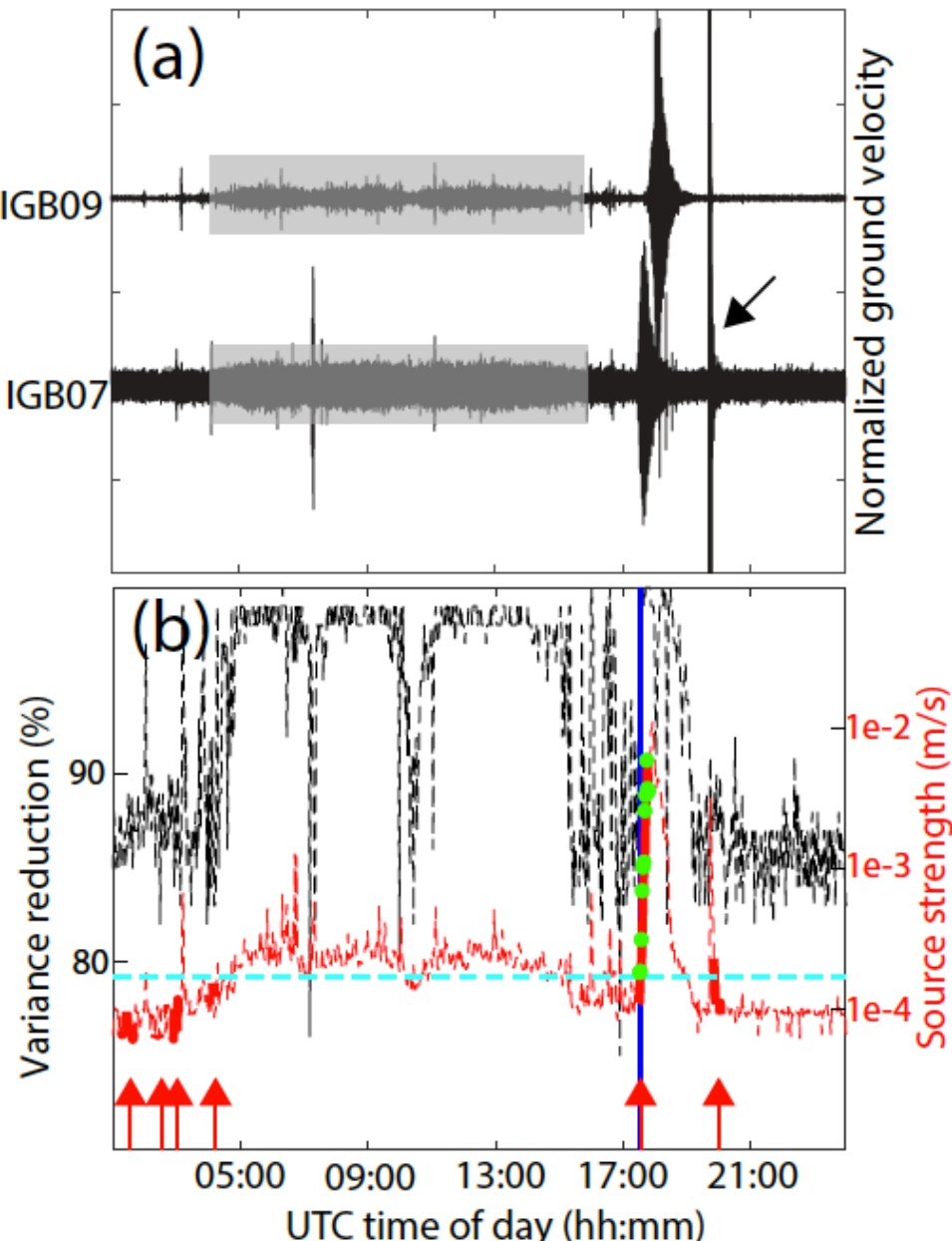

Figure 9: Results of the ASL approach. (a) Seismograms of stations IGB07 (in the Illgraben catchment) and IGB09 (in the Rhone Valley). Black arrow indicates high-frequency arrivals of the M6.2 2011 Fergana Valley earthquake in Kyrgyzstan. Grey boxes highlight increased amplitudes due to anthropogenic noise. (b) Variance reduction (black) and equivalent source strength (red). Thick solid red lines denote time instances when the best fit location lies in the Illgraben catchment (also indicated with red arrows), dashed lines denote the remaining time instances. Green dots represent times when three detection criteria were satisfied: (1) location in upper catchment, (2) variance reduction above 90 % and (3) source strength A0 above 1.7e-4 m/s. The blue vertical line marks 17:32, which we propose as the initial detection time of the debris flow.

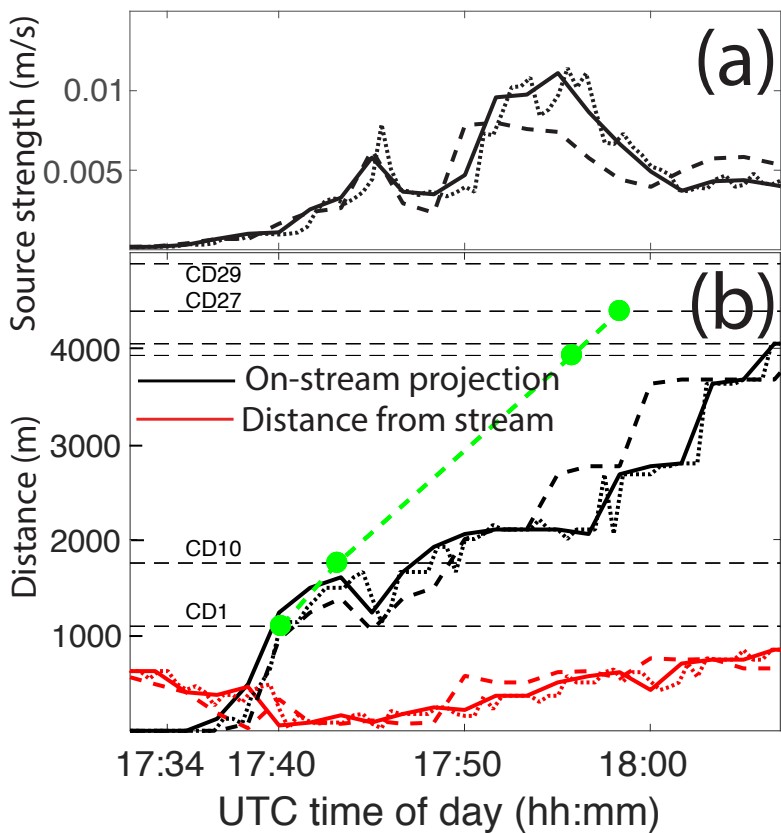

Figure 10: Source strength of the debris flow seismicity (a) and best-fit source locations (b). Solid line represents the ASL method applied to 100 s windows without site amplification correction. Dashed line represents the calculation with site amplification. Dotted line shows the results for 30 s windows, again without site amplification. In Panel (b), black lines show the best-fit locations projected onto the along-flow coordinate system of the stream with a manually picked point in the upper catchment indicating the channel head. Red lines show the shortest distance of the best-fit location from the streambed. Green dots connected by dashed lines indicate check dam arrival times (Labels of Check Dams 24 and 25 are omitted for clarity).

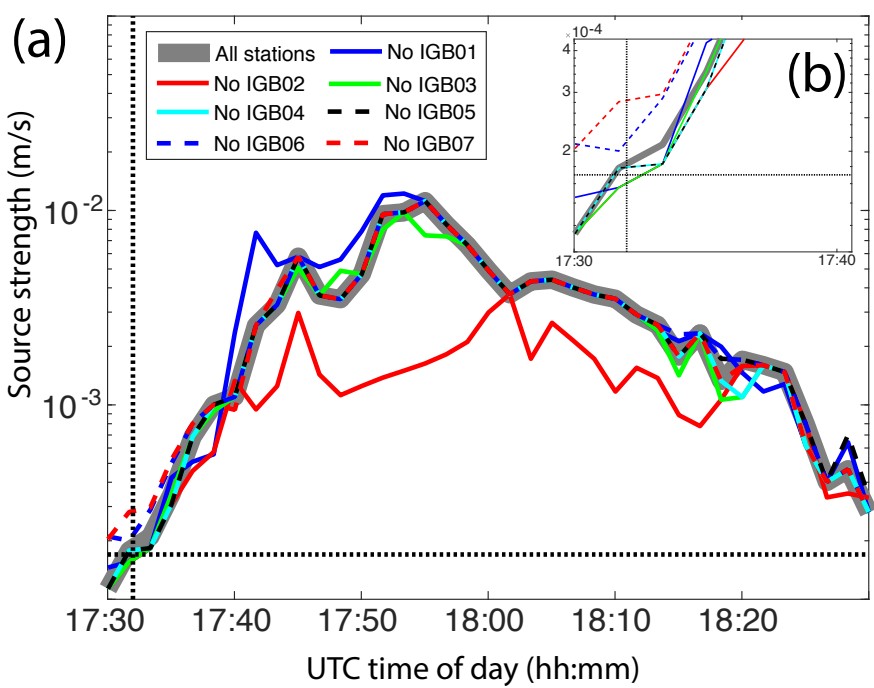

Figure 11: (a) Variations in calculated source strength $A_0$ as individual stations are removed. Vertical and horizontal bars indicate earliest detection time at 17:32 UTC and proposed threshold, respectively. (b) Zoom near earliest detection time at 17:32 UTC.

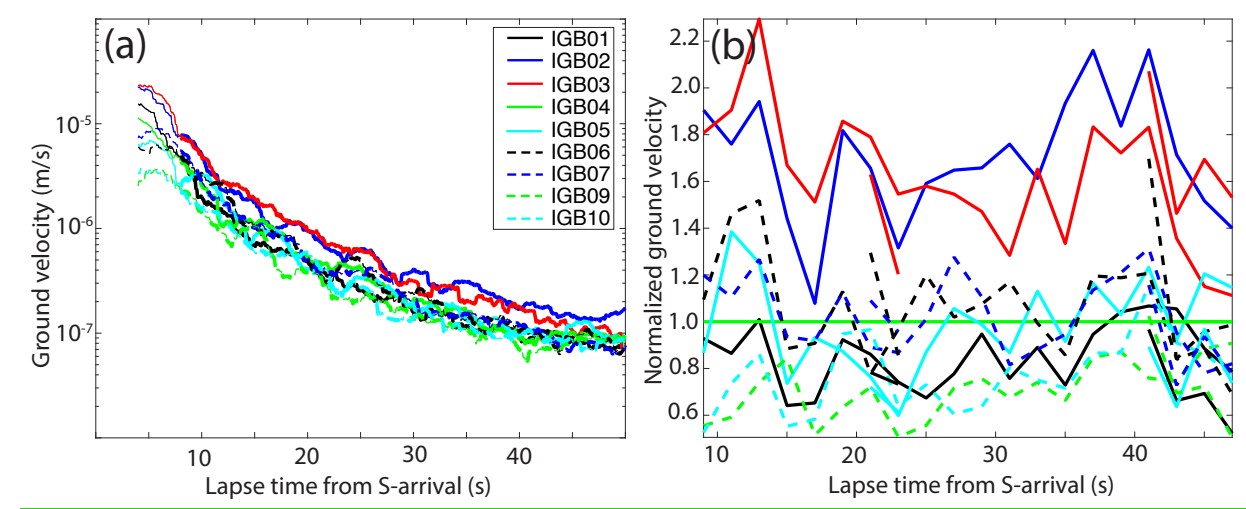

Figure 12: Coda amplification analysis. (a) S-wave coda envelopes of an M2.2 earthquake on 15 July 2011 (epicentral distance: 12 km). Bold lines indicate lapse times greater than twice the S-arrival time. (b) coda amplification with respect to station IGB04 using signals of the three earthquakes listed in Table 3.

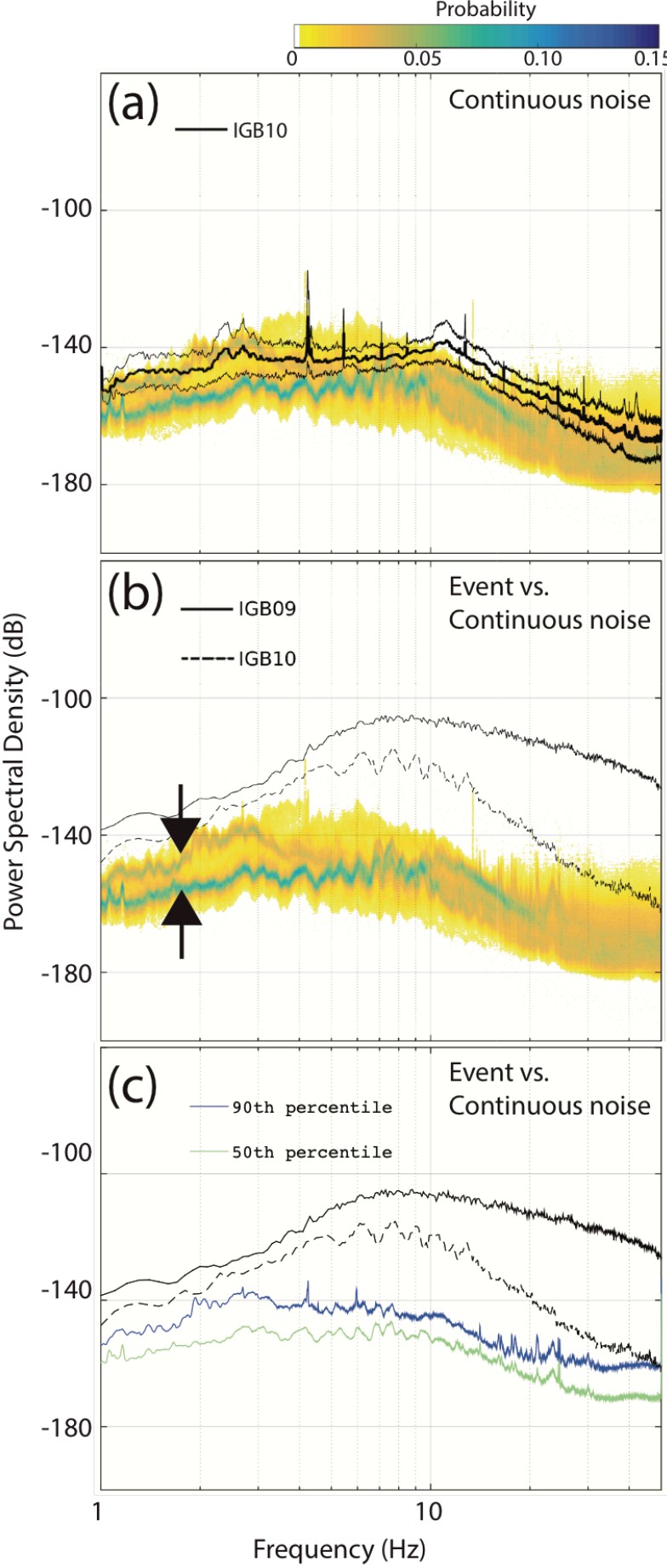

Figure 13: Debris flow spectra recorded at IGB09 and IGB10 together with probabilistic spectral representation of a 3-months noise time window recorded between 27 May and 16 July 2015 at station IGN01 ("PSD-PDF"). (a) Comparison between noise PSD-PDFs at IGN01 and noise mean and standard deviation at IGB10 for the day of the debris flow (thick and thin black lines, respectively). (b) Noise PSD-PDF of station IGN01 and debris flow spectra of IGB09 and IGB10. Black arrows point to the two PDF branches discussed in the main text. (c) Debris flow spectra at IGB09 and IGB10 with 50th and 90th percentile of noise PSD-PDF of IGN01. Note that at both stations (IGB09 and IGB10), the debris flow signal dominates the seismic noise measured at IGN01 over the entire shown frequency range (1-14 Hz).