# Peer review of "Testing Seismic Amplitude Source Location (ASL) for Fast Debris Flow Detection at Illgraben, Switzerland"

_Natural Hazards and Earth System Sciences, 2016_

## Referee Comment (RC1) · Anonymous Referee #1 · 16 Nov 2016

This study uses relative time-averaged seismic amplitudes recorded on a 9-station seismic network around the Illgraben torrent to detect and track a debris flow event and estimate it's relative seismic strength over time. There is an existing warning system and scientific observation system in place that uses observations on different instruments at check dams along the channel. The authors frame this study as a new approach to debris flow warning systems that overcomes some of the challenges of the existing warning system, for example, by allowing stations to be installed away from the channel in less difficult terrain. This is an important topic, as existing methods have limitations and don't take advantage of recent advances in seismology.

However, I do not feel that the paper is ready to be published. It's an interesting demonstration of a technique for tracking moving flows that is moderately successful at tracking a flow, but for a single flow that was known about beforehand and well-characterized

by other data sources. The technique applied is not new, yet the paper is framed as if it is, and they do not get into nearly enough detail regarding the limitations of the proposed method despite being unusually well suited to because of the trove of data from other existing instrumentation. In my opinion, in order to be publishable the paper needs to 1) be reframed in context of other studies that have used similar techniques and with a more modest/realistic approach to how these methods could be used in warning systems, 2) include more analysis to convince the reader of some of their claims and to illustrate limitations, which I detail below and in the specific comments, and 3) to include more specifics about the choices they made, why, and how those choices affect the results.

Regarding point one, the authors apply methods that have been used elsewhere for similar purposes (e.g. Kumagai et al. 2009, several references in Walsh et al. 2016 - refs at end), yet these studies are not even mentioned in the paper. The authors also do not seem to be aware that the volcano community has used acoustic flow monitors to detect lahars/debris flows relatively reliably for decades. Granted these systems have many of the same downsides as the system in place at Illgraben, but they still should be acknowledged. See links to some such studies listed at the end.

Regarding point 2, in other studies that have used similar methods, amplitude correction factors are derived for each seismic station to account for variation in site response. In the present study they skipped this step, stating that it was not needed because they were able to get a good solution without it. However, they did not convince me that it isn't necessary. Their method is not able to track the flow reliably after the initial few minutes (Fig 9), this is even more apparent in the supplemental movie. This could very well be because they do not correct for station response - some stations may have higher amplitudes than others in the frequency band used due to their specific site conditions, which would bias the location towards those particular stations. The authors should either add in the station corrections, or make a more convincing case demonstrating that it isn't necessary because it is hard to believe it isn't.

As for the decay fitting, the authors show that the variance reduction can be just as high when fitting noise as when fitting signal. This is not surprising because they are doing a grid search so they are basically rearranging the data points in many different distance configurations for each time step. The chances of having a decent fit somewhere are quite high. So they depend on the amplitude at the "source," A0, to differentiate noise from signal. However, I have personally tried something similar and found that A0 was extremely variable and highly dependent on the particular data points being used, especially if there weren't many data points close to the source. Excluding one or two outliers could drastically change the results. I would be much more convinced that their source strength estimations were providing a reliable way of discriminating signal from noise if there were some sensitivity analysis included. For example, the authors could use the jackknife technique to show how much A0 can vary by randomly excluding some of the data and redoing the fit many times. And/or they could run their algorithm for a long time period (they state they have 100 days of data on these stations...) and see how often they get false alarms based on the A0 thresholds they found for the known event.

Another thing that would be interesting to see and would be very relevant for the application of amplitude location methods to this type of seismic source (but may be beyond the scope of this small paper) is an analysis of what the solution looks like if there are two or more simultaneous sources, such as from multiple surges, or an elongated source. Can the technique differentiate between two sources? How far apart would they have to be? What does a pinpoint source look like using these methods (analogous to an array response function for array techniques)?

Regarding point 3, the authors leave out critical details (particularly on page 8) regarding why they made the choices they did in their implementation of these methods, what those choices were (actual values for things like seismic velocities, Q, window length etc.), and how varying those choices affected the solution. Without any information given, how do we know that they didn't just turn knobs until the method located the

event near where they knew it should be for part of the flow?

Specific comments:

P1-L18-19: It's not clear to me that the author's alternative solves this challenge. The seismometers still have to be installed in steep terrain and still have to be telemetered.

P1-L21-23: This implies that geophones aren't seismometers. What is different about the author's method is the algorithm, namely that it doesn't depend on the vibrations detected right next to the channel. This should be clarified.

P1-L24-27: Acoustic flow monitor-type detection systems also use time-averaged ground vibration amplitudes, just the ones right next to the channel, and do not rely on single station detections. The authors should clarify what is actually different about their method.

P1-L29: This implies that they applied this algorithm in real time, but from what I gather, they did this analysis long after the event occurred and was already characterized. For example, it is stated later on that the data was not even telemetered. Would the outcome have been so good in real time without prior knowledge of the existence of the event and with all the delays and complications of telemetry that the existing system already has to deal with?

P2-L22-23: The number quoted for maximum ground velocity surely isn't the highest of any debris flow ever, I would change this sentence to "ground motions of up to 2e-3 m/s have been observed..." Also, observable frequencies near the channel are often much higher than 100 Hz, acoustic flow monitoring systems often look at bands of several hundred Hz. For example, see Marcial et al. 1999.

P2-L29: I don't know if I agree that no reliable implementation for debris flows has been found. The authors describe one in the next few pages, and acoustic flow monitoring setups have been pretty reliable at volcanoes, though they certainly could be improved. The method proposed also requires site-specific parameter tuning (seismic

velocity structure, station amplification factors etc.), so that's still a factor. Also, the use of the term "single station detections" is misleading. To my knowledge, none of the existing methods are single station systems. They depend on time-delayed detections on multiple stations along the channel, including the method currently in place at Illgraben.

P3-L4-5: The phrasing of this sentence implies that moving the instruments away from the torrent decreases the influence of site effects on ground motion, I don't see why this would necessarily change anything regarding site effects, the new site will also have site effects. Increasing the distance also adds another challenge, path effects through an unknown subsurface structure.

P5-L29: If the stations had sample rates of 125 to 200 Hz, as stated above, you would only be able to see up to the nyquist frequencies of 63 or 100 Hz, respectively. . .the spectrogram shown in Fig 4 is from a station that Table 1 says was sampled only at 125 Hz, so we should only be able to see up to 63 Hz yet the spectrogram goes up to 100 Hz. This is not informative about the upper limit of the frequencies observed; there could be and probably are higher frequencies present.

P6-L8: The two papers referenced are about bedload in river flow, NOT debris flows, though the processes described may be similar between them seismically. Regardless, this should be made clear in the text to justify.

P6-L20-21: I'm not sure panel a is actually showing relative amplitudes as the text implies. I think it may be scaled to the maxes. The relative amplitudes don't look the same in panel b as where the red line is in panel a. For example, IGB09 looks lower in amplitude at the red line than IGB07 in panel a, but the opposite in panel b. Also, the Rhone Valley stations mentioned in the text don't have the highest amplitudes in panel a, other stations look just as high.

P7-L12: Signal coherence depends on how close the stations are to each other and the frequencies of interest, if the stations were actually in an array configuration designed for the frequencies of interest here, the signals very well could have been coherent.

The statements here imply that no debris flow signals are coherent ever.

P7-L18: Kumagai et al. 2009 should be referenced here. They used similar methods for essentially the exact same purpose you did, just on a much larger scale. Others have used these methods for debris flows and other surface flows, see Walsh et al. 2016 and references within.

P8-L10-11: Station corrections should not be neglected in my opinion, see main comments.

P8-L13-L25: Many critical details are missing here regarding why the authors made the choices they did, what those choices were (actual values), and how varying those choices affects the solution (see main comment above). More specifically, why did they decide to assume body waves? Body waves are not going to follow straight line paths, surface waves would (approximately). Some people argue that the strongest waves from surface flows are surface waves. Do the authors have evidence one way or the other here? Could using surface waves produce a similar result? Third, are they assuming P or S waves? Why? What velocity are they using? Why? Why did they choose 100 sec windows for amplitude average? What are the values of alpha and Q that they use that are "within the range expected for body waves near the surface of the earth"?

P9-L14-15: Since source strength is their best way to distinguish between noise and debris flow signals, a time series of source strength solutions should be shown on the movie and fig 6 and 7 in a similar way to how the seismic data is shown so we know how to interpret the solution at each point in time.

P9-L24-26: This statement lacks evidence. It could just as easily be because they aren't correcting your station amplitudes and that may inflict more of a bias on locations further downstream than upstream. Either show evidence for the statement here or show that the amplitude corrections really don't make a difference.

P10-L16: How do the authors exclude these stations to avoid affecting the decay fit?

P11-L16: It would be more convincing if the authors tried adding white noise comparable to what is sometimes seen, for example, during a storm, to the signals during the debris flow to see how it actually affects the decay fit scheme.

Figure 3: The geophone data looks processed in some way, are these envelopes of the amplitude data? Time averaged absolute values of amplitude? Why is the geophone signal flat before the arrival of the debris flow, is the instrument turned on by some sort of trigger? The label "geophone impulses" is vague in meaning. Please clarify these things either in the text (page 5) or in the caption, or both.

Fig 6 and 7: A0 for the fit at the initiation is lower than A0 for the noise window – this is confusing because the text implies that the A0 value was the main way they were able to differentiate between signal and noise (e.g., Fig 8)

Minor/editorial comments:

P2-L18: Debris flows do not always move so slowly, I would add a qualifier like "debris flows typically move at. . ."

P3-L2: The Arattano 1999 reference is missing from the reference list

P4-L5: What type of material are the slopes made of?

P4-L19: What is an "instrumented wall"?

P4-L20-24: It would be nice to have a map of this layout or photos to help visualize the setup.

P5-L12: Is this time local or UTC?

P9-L20: Are the authors projecting to the channel? Or is this the rate the best fitting location moves regardless of channel location?

P9-L24: lacks -> lags

Figure 1: Label the check dams on the map so we know which is which. Also, an inset map showing where Illgraben is in Switzerland would be helpful.

Figure 2 is not mentioned anywhere in the text.

Figure 9: Is distance referring to distance along the channel? Starting from where?

References:

Kumagai, H., Palacios, P., Maeda, T., Castillo, D., Nakano, M., 2009. Seismic tracking of lahars using tremor signals. J. Volcanol. Geotherm. Res. 183, 112–121.

Walsh, B., A.D. Jolly and J. N. Proctera, 2016, Seismic analysis of the 13 October 2012 Te Maari, New Zealand, lake breakout lahar: Insights into flow dynamics and the implications on mass flow monitoring, JVGR, vol 324, pg 144-155.

Links to some acoustic flow monitor papers:

Marcial et al. 1996, Instrumental Lahar Monitoring at Mount Pinatubo, from Fire and Mud, available at: https://pubs.usgs.gov/pinatubo/marcial/

https://pubs.usgs.gov/pinatubo/tungol/

https://volcanoes.usgs.gov/volcanoes/mount_rainier/mount_rainier_monitoring_99.html

http://www.sciencedirect.com/science/article/pii/S0377027300001517

---

## Referee Comment (RC2) · Anonymous Referee #2 · 22 Nov 2016

**RE:** NHESS 2016 321          Walter et al.  Rapid Detection location of debris flow at Illgraben, Switzerland

**Overview**

This paper presents an application of seismic signal analysis for detecting the debris flow initiation in not accessible sites. Therefore, it is well suited to NHESS. However, the submitted manuscript is not yet ready for publication. The writer in the two points below, shows two main deficiencies that have to be worked out.

1) As not all the readers are expert on seismic signal analysis, the  writer recommends a better and wider explanation of the method for determining debris flow location through the analysis of seismic data.

2) An analysis of the influence of distance of seismometers from the initiation site on their efficiency in detecting the debris flow triggering is needed.

**Specific comments**

Title

The proposed methodology for debris flow detection has been verified in an unique case, so that about title I propose the following:

A case of Rapid Detection location of debris flow at Illgraben, Switzerland

Abstract

The sentence at line 25 is understandable only after reading the entire paper. Please provide more understandable explanations.

Introduction

Moreover, what could it happen in the case of occurrence of other debris flows in the neighbouring areas? The proposed methodology could be able to identify the exact location of debris flow?

The authors should also consider this eventuality in the introduction and conclusions: just write some sentences that clarify this aspect.

**2 Illgraben debris flows**

page 4: debris flow initiation

The writer believes that abundant runoff production mobilizes the sediments laying in the main channel triggering the debris flows (runoff generated debris flows). This is the main triggering mechanism and it is very common in Alps (Berti and Simoni, 2005; Gregoretti and Dalla Fontana, 2008; Theule et al., 2012) as elsewhere (Cannon et al., 2008; Coe et al., 2008; Hurlimann et al., 2014). Recent studies (Kean et al., 2012; Rengers et al., 2016; Gregoretti et al., 2016) have shown that the hydrological response of steep slopes is very fast and provides large peaks flows that are able to mobilize large quantities of sediments triggering debris flow phenomena. Main source of sediments are the steep lateral slopes. Sediments delivered to the main channel could also obstructing it, forming a dam (Costa and Shuster, 1988; Clague and Evans, 1994). In this last case, the failure/erosion of the formed dam due to runoff after thunderstorm cause a large solid-liquid wave. This is a particular case of the runoff generated debris flow and could also be seen in the video of the debris flow occurred at Acquabona the 12th of June 1997 (see web page of Matteo Berti, University of Bologna) where runoff impact a debris deposit, originated by banks failure that obstructs the channel.

I suggest the authors to adapt the description above in the explanation of debris flow occurrence at Illgraben.

**3 Seismic data**

I suggest to eliminate IGB8 and renumbering the following seismometers.

Panel A of Figure 5 shows that the amplitude of signal corresponding to the green bar is very large for IGB07 while this does not appear in panel C. What about the difference between normalized ground velocity and scaled ground velocity? Some explanations in the text is due.

**4 Detection and Location Scheme**

Points, 2, 3 and 4 at pages 6 and 7 look like statements rather than demands. They, together the explanations points to points below, could be presented at the beginning justifying the proposed approach.

The writer does not understand the computation of debris flow location through decay fitting. The analysis of the measured signal amplitude shows the exact moment of debris flow occurrence due to the high increase of the measured signal amplitude. About equation (1) $A_i$ is a data and r is the unknown quantity. How $A_O$ can be determined? Moreover, some more explanations on the matching between RMS distribution and eq. (1) predictions could help the reader.

Equation (3): What is it fit? The RMS? This should explained because most of readers are not expert on the analysis and use of seismic data.

5 Results: Seismic Noise....

Figures 6 and 7. The upper green triangle seems IGB10 rather than IGB9. Moreover, I suggest to label the black triangles corresponding to IGB01, IGB02 and IGB03.

Figure 6 caption. What about black cross? I do not see them.

Line 31 of page 9. The writer does not understand the distance from variance reduction maximum: in the caption of Figure 9 there is no information about distance from variance reduction maximum as the ordinate of the panel B. Moreover, add the label 1000 and 3000 in the vertical axis.

6 Discussion: detectability and background noise and Conclusions

Please add some comment about the influence of the distance of the seismometer station from the debris flow occurrence location. Panel C of Figure 5 shows that only signal from IGB01, IGB02 and IGB03 are marked. Were also the other signal from IGB04-IGB10 used for computing decay fitting? In the case that station IGB01, IGB02 and IGB03 are missing the results from seismic data are the same? Please add some comment explanation.

**Technical corrections**

CD9 is missing in Figure 1

line 9 of page 8: demands instead of demand?

Berti, M., and A. Simoni (2005), Experimental evidences and numerical modelling of debris flow initiated by channel runoff, *Landslide*, 2, 171--182.

Cannon, S., Gartner J.E., Wilson, R.C., Bowers, J.C., Laber, J.L. 2008. Storm rainfall conditions for floods and debris flows from recently burned areas in Southwestern Colorado and Southern California. *Geomorphology*, 96, 250-269.

Coe, J.A., Kinner D.A., Godt, J.W., 2008. Initiation conditions for debris flows generated by runoff at Chalk Cliffs, central Colorado. *Geomorphology*, 96, 270-297.

Clague, J. and Evans, G. (1994). Formation and failure of natural dams in the Canadian Cordillera,*Geological Survey of Canada Bulletin* 464

Costa, J.E., and Shuster R.L. (1988) The formation and failure of natural dams.,*Geological Society of America Bulletin*, v 100, 1054-1068.

Gregoretti, C., Dalla Fontana G., 2008. The triggering of debris flow due to channel-bed failure debris flow in some alpine headwater basins of the Dolomites: analyses of critical runoff. *Hydrological Processes*. 22, 2248-2263.

Gregoretti C., Degetto M., Bernard M., Crucil, G., Pimazzoni A., De Vido G., Berti M., Simoni A. Lanzoni S. Runoff of small rocky headwater catchments: Field observations and hydrological modeling. *Water Resources Research*. 52(8) doi: 10.1002/2016WR018675

Hurlimann M., Abanco C., Moya, J., Vilajosana I. (2014). Results and experiences gathered at the Rebaixader debris-flow monitoring site, Central Pyrenees, Spain. *Landslides*. doi:10.1007/s10346-013-0452-y 161-175

Kean J.W., Staley D.M., Leeper R.J., Schmidt K.M., Gartner J.E. (2012). A low-cost method to measure the timing of postfire flash floods and debris flows relative to rainfall. *Water Resources Research*, 48, W05516, doi:10.1029/2011WR011460

Rengers, F.K., L.A. McGuire, J.W. Kean and D.E. Hobley (2016), Model simnulations of flood and debris flow timing in steep cachments after wildfire, *Water Resources Research*, 52, doi:10.1029/2015WR018176.

Theule, J.I., Liebault, F., Loye, A., Laigle, D., and Jaboyedoff, M., 2012. Sediment budget monitoring of debris flow and bedload transport in the Manival Torrent, SE France. *Natural Hazard Earth Sciences*, 12, 731-749.

---

## Author Comment (AC1) · 20 Dec 2016

Dear Editor,

We thank you for considering our submitted manuscript and the referees for providing very constructive and critical feedback. Both referees listed numerous references, which we were unaware of and consequently did not cite or discuss. In a revised manuscript version, we intend to rectify this shortcoming. In particular, Referee #1 pointed out that our proposed amplitude decay location has been implemented for detection and monitoring of lahars. Consequently, we will modify the scope of our manuscript and frame it as a study on implementation and testing against ground truth in an early warning context. Similarly, we will review details on debris flow initiation as suggested by Referee #2.

[Figure]

Furthermore, we intend to perform additional analysis steps as suggested by Referee #1. Specifically, we will investigate site effects and the robustness of the location results, in particular the sensitivity of the source strength parameter.

Given these modifications and the ones discussed below, we are confident that our manuscript will undergo significant improvement and request the opportunity to resubmit.

Fabian Walter (on behalf of the author team)

---

## Author Comment (AC2) · 20 Dec 2016

Reviewer: 1) reframing "in context of other studies that have used similar techniques and with a more modest/realistic approach to how these methods could be used in warning systems"

Authors: Admittedly, we were unaware of these previous studies. Discussing them will naturally lead to a more modest/realistic presentation of our work as an implementation of an existing technique and an exploratory exercise of its suitability for early warning purposes.

R: 2) include more analysis to convince the reader of some of their claims and to illustrate limitations, which I detail below and in the specific comments

[Figure]

A: We will describe and carry out additional analysis steps. The referee emphasized the need for investigating site effects. Initially we had refrained from this step, because we were looking for a workflow, which is straightforward to apply and highly portable to other debris flow catchments. However, with the hope to improve our location quality, we will investigate coda amplification factors of regional and local earthquakes. For this task we have identified about three local and regional earthquakes, which show good signal-to-noise-ratios on a nearby permanent station of the Swiss Seismological Service.

R: 3) include more specifics about the choices they made, why, and how those choices affect the results

A: This information will be provided.

Specific Comments

R: P1-L18-19: It's not clear to me that the author's alternative solves this challenge. The seismometers still have to be installed in steep terrain and still have to be telemetered.

A: We will explain more clearly how seismometers provide flexibility for installations.

R: P1-L21-23: This implies that geophones aren't seismometers. What is different about the author's method is the algorithm, namely that it doesn't depend on the vibrations detected right next to the channel. This should be clarified.

A: We will adapt this explanation and discuss geophone installations in and near torrent channels in more detail.

R: P1-L24-27: Acoustic flow monitor-type detection systems also use time-averaged ground vibration amplitudes, just the ones right next to the channel, and do not rely on single station detections. The authors should clarify what is actually different about their method.

A: This will be discussed in detail using the references, which this referee provided.

R: P1-L29: This implies that they applied this algorithm in real time, but from what I gather, they did this analysis long after the event occurred and was already characterized. For example, it is stated later on that the data was not even telemetered. Would the outcome have been so good in real time without prior knowledge of the existence of the event and with all the delays and complications of telemetry that the existing system already has to deal with?

A: We will make it clearer that we did our analysis offline in retrospect and discuss challenges for implementation in real time.

R: P2-L22-23: The number quoted for maximum ground velocity surely isn't the highest of any debris flow ever, I would change this sentence to "ground motions of up to 2e-3 m/s have been observed: : :" Also, observable frequencies near the channel are often much higher than 100 Hz, acoustic flow monitoring systems often look at bands of several hundred Hz. For example, see Marcial et al. 1999.

A: These suggestions will be followed.

R: P2-L29: I don't know if I agree that no reliable implementation for debris flows has been found. The authors describe one in the next few pages, and acoustic flow monitoring setups have been pretty reliable at volcanoes, though they certainly could be improved. The method proposed also requires site-specific parameter tuning (seismic velocity structure, station amplification factors etc.), so that's still a factor. Also, the use of the term "single station detections" is misleading. To my knowledge, none of the existing methods are single station systems. They depend on time-delayed detections on multiple stations along the channel, including the method currently in place at Illgraben.

A: As stated above, we will change the scope of our paper and revise/extend our description of existing early warning systems.

R: P3-L4-5: The phrasing of this sentence implies that moving the instruments away

from the torrent decreases the influence of site effects on ground motion, I don't see why this would necessarily change anything regarding site effects, the new site will also have site effects. Increasing the distance also adds another challenge, path effects through an unknown subsurface structure.

A: This part will be rephrased and likely change after our site effect analysis.

R: P5-L29: If the stations had sample rates of 125 to 200 Hz, as stated above, you would only be able to see up to the nyquist frequencies of 63 or 100 Hz, respectively: : :the spectrogram shown in Fig 4 is from a station that Table 1 says was sampled only at 125 Hz, so we should only be able to see up to 63 Hz yet the spectrogram goes up to 100 Hz. This is not informative about the upper limit of the frequencies observed; there could be and probably are higher frequencies present.

A: We will change the axis limits of this figure.

R: P6-L8: The two papers referenced are about bedload in river flow, NOT debris flows, though the processes described may be similar between them seismically. Regardless, this should be made clear in the text to justify.

A: Will be changed accordingly.

R: P6-L20-21: I'm not sure panel a is actually showing relative amplitudes as the text implies. I think it may be scaled to the maxes. The relative amplitudes don't look the same in panel b as where the red line is in panel a. For example, IGB09 looks lower in amplitude at the red line than IGB07 in panel a, but the opposite in panel b. Also, the Rhone Valley stations mentioned in the text don't have the highest amplitudes in panel a, other stations look just as high.

A: The referee is right: Panel A shows normalized amplitudes. We will make this clear in the caption.

R: P7-L12: Signal coherence depends on how close the stations are to each other and the frequencies of interest, if the stations were actually in an array configuration

designed for the frequencies of interest here, the signals very well could have been coherent.

A: This statement was not intended to be general but was made considering the network configuration of our study. We will clarify this.

R: P7-L18: Kumagai et al. 2009 should be referenced here. They used similar methods for essentially the exact same purpose you did, just on a much larger scale. Others have used these methods for debris flows and other surface flows, see Walsh et al. 2016 and references within.

A: We will include these references.

R: P8-L10-11: Station corrections should not be neglected in my opinion, see main comments.

A: This will be analyzed with earthquake data (see above).

R: P8-L13-L25: Many critical details are missing here regarding why the authors made the choices they did, what those choices were (actual values), and how varying those choices affects the solution (see main comment above). More specifically, why did they decide to assume body waves? Body waves are not going to follow straight line paths, surface waves would (approximately). Some people argue that the strongest waves from surface flows are surface waves. Do the authors have evidence one way or the other here? Could using surface waves produce a similar result? Third, are they assuming P or S waves? Why? What velocity are they using? Why? Why did they choose 100 sec windows for amplitude average? What are the values of alpha and Q that they use that are "within the range expected for body waves near the surface of the earth"?

A: We had outsourced most of these descriptions to Burtin et al. (2013), but we can certainly include them here.

R: P9-L14-15: Since source strength is their best way to distinguish between noise and

debris flow signals, a time series of source strength solutions should be shown on the movie and fig 6 and 7 in a similar way to how the seismic data is shown so we know how to interpret the solution at each point in time.

A: Rather than producing a single parameter, such as source strength, to identify debris flow signals, our approach yields a set of discriminators: source strength, location and fit quality. We will make this clearer and also include a time series of source strength in the movie and figures as requested.

R: P9-L24-26: This statement lacks evidence. It could just as easily be because they aren't correcting your station amplitudes and that may inflict more of a bias on locations further downstream than upstream. Either show evidence for the statement here or show that the amplitude corrections really don't make a difference.

A: We expect the site effect analysis to clarify this point.

R: P10-L16: How do the authors exclude these stations to avoid affecting the decay fit?

A: The point of the noise analysis was in fact to show that station exclusion is not necessary. We will state this specifically.

R: P11-L16: It would be more convincing if the authors tried adding white noise comparable to what is sometimes seen, for example, during a storm, to the signals during the debris flow to see how it actually affects the decay fit scheme.

A: Our probability-based noise analysis was meant to show that noise events with PSD comparable to our debris flow event occur rarely or never. We are unaware of the fact that storms map into white noise on a seismic recording, but we will investigate white noise effects with numerical tests.

R: Figure 3: The geophone data looks processed in some way, are these envelopes of the amplitude data? Time averaged absolute values of amplitude? Why is the geophone signal flat before the arrival of the debris flow, is the instrument turned on

by some sort of trigger? The label "geophone impulses" is vague in meaning. Please clarify these things either in the text (page 5) or in the caption, or both.

A: Will be clarified.

R: Fig 6 and 7: A0 for the fit at the initiation is lower than A0 for the noise window – this is confusing because the text implies that the A0 value was the main way they were able to differentiate between signal and noise (e.g., Fig 8)

A: We will clarify that A0 by itself is not robust for this very reason. As shown in Figure 8, location has to be taken into account, as well.
* * *

---

## Author Comment (AC3) · 20 Dec 2016

Reviewer: 1) As not all the readers are expert on seismic signal analysis, the writer recommends a better and wider explanation of the method for determining debris flow location through the analysis of seismic data.

Authors: We will include these details.

R: 2) An analysis of the influence of distance of seismometers from the initiation site on their efficiency in detecting the debris flow triggering is needed.

A: We will discuss this in more detail and include results of our site effect analysis.

Specific Comments

R: The proposed methodology for debris flow detection has been verified in an unique

case, so that about title I propose the following: A case of Rapid Detection location of debris flow at Illgraben, Switzerland

A: As our method has been previously applied to lahars, we will change the context of our manuscript and consequently the title (see Referee #1: General Comments).

R: Abstract: The sentence at line 25 is understandable only after reading the entire paper. Please provide more understandable explanations.

A: Explanations will be provided.

R: Introduction: Moreover, what could it happen in the case of occurrence of other debris flows in the neighbouring areas? The proposed methodology could be able to identify the exact location of debris flow? The authors should also consider this eventuality in the introduction and conclusions: just write some sentences that clarify this aspect.

A: A seismic signal of such a remote debris flow is clearly distinguishable by its amplitude decay throughout our network. We will clarify and explain this.

R: Page 4: debris flow initiation: [...] I suggest the authors to adapt the description above in the explanation of debris flow occurrence at Illgraben.

A: We thank the referee for these references and will discuss them in the manuscript.

R: Seismic data: I suggest to eliminate IGB8 and renumbering the following seismometers.

A: We prefer leaving the station numbering to keep consistency with previous publications on this data set.

R: Seismic data: Panel A of Figure 5 shows that the amplitude of signal corresponding to the green bar is very large for IGB07 while this does not appear in panel C. What about the difference between normalized ground velocity and scaled ground velocity? Some explanations in the text is due.

[Figure]

A: We will extend the caption (see equivalent comment by Referee #1).

R: Detection and location scheme: Points, 2, 3 and 4 at pages 6 and 7 look like statements rather than demands. They, together the explanations points to points below, could be presented at the beginning justifying the proposed approach.

A: We will move these points to an earlier part of the manuscript.

R: Detection and location scheme: The writer does not understand the computation of debris flow location through decay fitting. The analysis of the measured signal amplitude shows the exact moment of debris flow occurrence due to the high increase of the measured signal amplitude. About equation (1) Ai is a data and r is the unknown quantity. How AO can be determined? Moreover, some more explanations on the matching between RMS distribution and eq. (1) predictions could help the reader.

A: We will include more details on the location calculation and grid search.

R: Detection and location scheme: Equation (3): What is it fit? The RMS? This should explained because most of readers are not expert on the analysis and use of seismic data.

A: We will include the requested details.

R: Results: Seismic noise ... Figures 6 and 7. The upper green triangle seems IGB10 rather than IGB9. Moreover, I suggest to label the black triangles corresponding to IGB01, IGB02 and IGB03.

A: Sorry for the mislabeling. We will correct this.

R: Results: Seismic noise ... Figure 6 caption. What about black cross? I do not see them.

A: We will make the black cross more obvious.

R: Results: Seismic noise ... Line 31 of page 9. The writer does not understand

the distance from variance reduction maximum: in the caption of Figure 9 there is no information about distance from variance reduction maximum as the ordinate of the panel B. Moreover, add the label 1000 and 3000 in the vertical axis.

A: We will add explanations and labels accordingly.

R: Discussion: detectability and background noise and Conclusion: Please add some comment about the influence of the distance of the seismometer station from the debris flow occurrence location. Panel C of Figure 5 shows that only signal from IGB01, IGB02 and IGB03 are marked. Were also the other signal from IGB04-IGB10 used for computing decay fitting? In the case that station IGB01, IGB02 and IGB03 are missing the results from seismic data are the same? Please add some comment explanation.

A: We intend to include another panel in Figure 5, which shows the debris flow seismograms at a different time from Panel C. This should make clear how amplitudes at different station subsets participate in the location grid search at different times.
* * *

---

## Author Response (AR1)

Dear Editor,

We have revised our manuscript according to the remarks and suggestions of the two referees. A point-by-point explanation of our responses to the referees' comments can be found below. However, as our manuscript changed substantially, we list the main new features.

We no longer frame our study as a new method of debris flow detection. Instead, we describe previous applications of this method and how we investigate its suitability for automated detections and early warning systems in a catchment, where state-of-the-art alarms systems and measurements exist for ground truth comparison. Accordingly, we now call our method "amplitude source location (ASL)" in agreement with previous studies.

The current manuscript version is more modest on the capabilities, which the ASL method may afford, and our early warning time estimates are more conservative. Previously implemented early-warning systems using seismology are acknowledged and we removed statements that suggest an absolute superiority of our method.

We now document processing of a longer time series than the original 19 hours and we include a discussion of false detections and how in future automated processing they can be caught. This lead to the modification and extension of several figures and a new figure showing the effect of single station removal is now shown. Moreover, we present two other debris flow seismograms and offer a discussion on how the ASL technique may perform on other events.

Furthermore, we rewrote the introduction to better and more comprehensively describe the use of seismic monitoring in debris flow research. We also discuss different trigger mechanisms of debris flows.

In the annotated manuscript, our changes are highlighted. However, we did not highlight changes in the introduction and the discussion on early warning suitability, which were completely rewritten. We also point out that following the referees' suggestions we made various changes to the figures and supplemental material. Most shown time series were extended to include the end of day 19 July 2011 and Figure 9 now shows IGB07 instead of IGB01.

Finally, we edited the animated movie and included a new movie in 3D view for illustrative purposes.

Fabian Walter
(on behalf of the author team)

POINT-BY-POINT REPLY

Referee #1

*Reviewer: 1) reframing "in context of other studies that have used similar techniques and with a more modest/realistic approach to how these methods could be used in warning systems"*

Authors: Following this suggestion, we restructured the introduction and changed other parts in the manuscript. Now we discuss previous studies using the ASL technique and our scope changed towards testing the suitability of this technique for early warning given the ground-truth data available at Illgraben.

*R: 2) include more analysis to convince the reader of some of their claims and to illustrate limitations, which I detail below and in the specific comments*

A: As described below, we performed a site amplification analysis, we tested location robustness by removing individual stations and we checked the performance when using smaller time windows.

*R: 2) an analysis of what the solution looks like if there are two or more simultaneous sources, such as from multiple surges, or an elongated source*

A: This point is clearly relevant to monitoring and it made us also look at the 13 July 2011 event, which we now discuss and present in Figure 4. As documented in Burtin et al. (2013), this event consisted of several flow pulses and concurrent rock fall events. This may indeed explain why our ASL processing was unsuccessful. On the other hand, tests on the effect of multiple sources, which the referee may have had in mind, would require calculation of synthetic seismograms and subsequent superposition to simulate a range of different scenarios. We feel that this reaches a complexity, which no longer fits into our current study.

*R: 3) include more specifics about the choices they made, why, and how those choices affect the results*

A: The specifications are now included.

Specific Comments

*R: P1-L18-19: It's not clear to me that the author's alternative solves this challenge. The seismometers still have to be installed in steep terrain and still have to be telemetered.*

A: We explain that the advantage of moving stations away from a channel is additional flexibility on installation sites.

*R: P1-L21-23: This implies that geophones aren't seismometers. What is different about the author's method is the algorithm, namely that it doesn't depend on the vibrations detected right next to the channel. This should be clarified.*

A: In the introduction, we changed and expanded our discussion of ground vibration sensing, which we generally refer to as seismic methods. Although we briefly mention distinct

frequency sensitivities of seismometers and geophones, we prefer not emphasizing the differences between acoustic sensors, geophones and seismometers, because even though all terms are widely used, there does not seem to be a clear, standard definition that distinguishes between them.

*R: P1-L24-27: Acoustic flow monitor-type detection systems also use time-averaged ground vibration amplitudes, just the ones right next to the channel, and do not rely on single station detections. The authors should clarify what is actually different about their method.*

A: The new introduction aims to better explain these differences.

*R: P1-L29: This implies that they applied this algorithm in real time, but from what I gather, they did this analysis long after the event occurred and was already characterized. For example, it is stated later on that the data was not even telemetered. Would the outcome have been so good in real time without prior knowledge of the existence of the event and with all the delays and complications of telemetry that the existing system already has to deal with?*

A: In the introduction we now make it clear that we use archived seismic data. Also, we emphasize that our work bolsters suitability of the ASL method for real-time debris flow detection, but more data are clearly needed for tuning parameters, such that the method can be used in an operational setting.

*R: P2-L22-23: The number quoted for maximum ground velocity surely isn't the highest of any debris flow ever, I would change this sentence to "ground motions of up to 2e-3 m/s have been observed: : :" Also, observable frequencies near the channel are often much higher than 100 Hz, acoustic flow monitoring systems often look at bands of several hundred Hz. For example, see Marcial et al. 1999.*

A: Changed.

*R: P2-L29: I don't know if I agree that no reliable implementation for debris flows has been found. The authors describe one in the next few pages, and acoustic flow monitoring setups have been pretty reliable at volcanoes, though they certainly could be improved. The method proposed also requires site-specific parameter tuning (seismic velocity structure, station amplification factors etc.), so that's still a factor. Also, the use of the term "single station detections" is misleading. To my knowledge, none of the existing methods are single station systems. They depend on time-delayed detections on multiple stations along the channel, including the method currently in place at Illgraben.*

A: Given the updated reference list, we now acknowledge other detection methods, which can certainly be called reliable and we acknowledge that our approach needs detector tuning as well. Also, we avoid the term "single-station methods" and now state that we explore how the ASL method can improve existing early warning schemes.

*R: P3-L4-5: The phrasing of this sentence implies that moving the instruments away from the torrent decreases the influence of site effects on ground motion, I don't see why this would necessarily change anything regarding site effects, the new site will also have site effects. Increasing the distance also adds another challenge, path effects through*

*an unknown subsurface structure.*

A: We changed the text to avoid leaving the impression that moving sensors away from the torrent could mitigate site effects. However, this does allow some control on site effects, which we now state.

*R: P5-L29: If the stations had sample rates of 125 to 200 Hz, as stated above, you would only be able to see up to the nyquist frequencies of 63 or 100 Hz, respectively: : :the spectrogram shown in Fig 4 is from a station that Table 1 says was sampled only at 125 Hz, so we should only be able to see up to 63 Hz yet the spectrogram goes up to 100 Hz. This is not informative about the upper limit of the frequencies observed; there could be and probably are higher frequencies present.*

A: Thank you for catching this. We noticed that the information in Table 1 is wrong: IGB02 sampled at 200 Hz and IGB09 and IGN10 sampled at 125 Hz. This was corrected and Figure 4 (now Figure 5) was left as is.

*R: P6-L8: The two papers referenced are about bedload in river flow, NOT debris flows, though the processes described may be similar between them seismically. Regardless, this should be made clear in the text to justify.*

A: We no longer suggest that the references are exclusively associated with debris flows.

*R: P6-L20-21: I'm not sure panel a is actually showing relative amplitudes as the text implies. I think it may be scaled to the maxes. The relative amplitudes don't look the same in panel b as where the red line is in panel a. For example, IGB09 looks lower in amplitude at the red line than IGB07 in panel a, but the opposite in panel b. Also, the Rhone Valley stations mentioned in the text don't have the highest amplitudes in panel a, other stations look just as high.*

A: The referee is right: Panel a shows normalized amplitudes with respect to the maximum of each trace and hence, the relative amplitudes between traces in Panel a have no meaning. We now make this clear in the caption of the updated figure.

*R: P7-L12: Signal coherence depends on how close the stations are to each other and the frequencies of interest, if the stations were actually in an array configuration designed for the frequencies of interest here, the signals very well could have been coherent.*

A: This statement was not intended to be general but was made considering a network configuration covering a larger area. We clarified this.

*R: P7-L18: Kumagai et al. 2009 should be referenced here. They used similar methods for essentially the exact same purpose you did, just on a much larger scale. Others have used these methods for debris flows and other surface flows, see Walsh et al. 2016 and references within.*

A: We included these references.

*R: P8-L10-11: Station corrections should not be neglected in my opinion, see main comments.*

A: This was analyzed with earthquake data, and we now include an extra section on this topic.

*R: P8-L13-L25: Many critical details are missing here regarding why the authors made the choices they did, what those choices were (actual values), and how varying those choices affects the solution (see main comment above). More specifically, why did they decide to assume body waves? Body waves are not going to follow straight line paths, surface waves would (approximately). Some people argue that the strongest waves from surface flows are surface waves. Do the authors have evidence one way or the other here? Could using surface waves produce a similar result? Third, are they assuming P or S waves? Why? What velocity are they using? Why? Why did they choose 100 sec windows for amplitude average? What are the values of alpha and Q that they use that are "within the range expected for body waves near the surface of the earth"?*

A: We now specify the grid search parameter range. The surface wave formulation also provided reasonable results, and a systematic investigation of this matter would be interesting but is referred to a future study. The 100 sec window was chosen to obtain a smooth propagation of the debris flow front. However, in view of early warning, this window size should be decreased and we now include a calculation, which shows that this is possible (Figure 10).

*R: P9-L14-15: Since source strength is their best way to distinguish between noise and debris flow signals, a time series of source strength solutions should be shown on the movie and fig 6 and 7 in a similar way to how the seismic data is shown so we know how to interpret the solution at each point in time.*

A: Detection with the ASL cannot rely on source strength, variance reduction or source location independently. These parameters have to be combined. We now make this clearer and discuss detection conditions in detail.

*R: P9-L24-26: This statement lacks evidence. It could just as easily be because they aren't correcting your station amplitudes and that may inflict more of a bias on locations further downstream than upstream. Either show evidence for the statement here or show that the amplitude corrections really don't make a difference.*

A: The site effects seem to have a minor effect, which we show and discuss in a new section.

*R: P10-L16: How do the authors exclude these stations to avoid affecting the decay fit?*

A: The point of the noise analysis was in fact to show that station exclusion is not necessary. For this reason, we analyzed the noise on a valley station. We now state this specifically.

*R: P11-L16: It would be more convincing if the authors tried adding white noise comparable to what is sometimes seen, for example, during a storm, to the signals during the debris flow to see how it actually affects the decay fit scheme.*

A: Our probability-based noise analysis was meant to show that noise events with PSD comparable to our debris flow event occur rarely or never. We ended up deciding against tests with artificial noise, because it is difficult to choose the correct frequency spectrum of a

potential noise signature. Furthermore, the noise PSD on stations affected by anthropogenic sources seems to have a "binary" character as shown in Figure 13. A systematic study of noise variations using long-term continuous records from all sensors could be done in future investigations, but we feel it is beyond the scope of the present study, especially since the current data set includes large gaps on several stations.

*R: Figure 3: The geophone data looks processed in some way, are these envelopes of the amplitude data? Time averaged absolute values of amplitude? Why is the geophone signal flat before the arrival of the debris flow, is the instrument turned on by some sort of trigger? The label "geophone impulses" is vague in meaning. Please clarify these things either in the text (page 5) or in the caption, or both.*

A: We included technical details on the geophone data processing and additional references on this topic.

*R: Fig 6 and 7: A0 for the fit at the initiation is lower than A0 for the noise window – this is confusing because the text implies that the A0 value was the main way they were able to differentiate between signal and noise (e.g., Fig 8)*

A: We extended our discussion on triggering thresholds clarifying that A0 by itself is not robust for this very reason. As shown in Figure 9, location has to be taken into account, as well.

Minor/editorial comments:

*P2-L18: Debris flows do not always move so slowly, I would add a qualifier like "debris flows typically move at: : :"*

Done.

*P3-L2: The Arattano 1999 reference is missing from the reference list*

We included the reference.

*P4-L5: What type of material are the slopes made of?*

We now include information and a reference on this.

*P4-L19: What is an "instrumented wall"?*

This is now explained.

*P4-L20-24: It would be nice to have a map of this layout or photos to help visualize the setup.*

As these measurements are not in the focus of this study, we decided against an extra figure to document the setup. We do include references, where such figures can be found.

*P5-L12: Is this time local or UTC?*

Yes, we specify this on all time axes now.

*P9-L20: Are the authors projecting to the channel? Or is this the rate the best fitting location moves regardless of channel location?*

We now explain that we project onto the channel.

*P9-L24: lacks -> lags*

Changed.

*Figure 1: Label the check dams on the map so we know which is which. Also, an inset map showing where Illgraben is in Switzerland would be helpful.*

Done.

*Figure 2 is not mentioned anywhere in the text.*

Now mentioned when introducing Illgraben debris flows.

*Figure 9: Is distance referring to distance along the channel? Starting from where?*

Distance is either shortest distance of the best-fit location to the channel or along channel starting from a manually picked point in the catchment. Now specified.

Referee #2:

*Reviewer: 1) As not all the readers are expert on seismic signal analysis, the writer recommends a better and wider explanation of the method for determining debris flow location through the analysis of seismic data.*

Authors: We included these details and wider explanations of the ASL method in the introduction.

*R: 2) An analysis of the influence of distance of seismometers from the initiation site on their efficiency in detecting the debris flow triggering is needed.*

A: We discussed the relation between amplitude and distance in more detail and how all station information is used. A new panel in Figure 6 aims at clarifying this.

Specific Comments

*R: The proposed methodology for debris flow detection has been verified in an unique case, so that about title I propose the following: A case of Rapid Detection location of debris flow at Illgraben, Switzerland*

A: We changed the title to: Testing Seismic Amplitude Source Location (ASL) for Rapid Debris Flow Detection at Illgraben, Switzerland.

*R: Abstract: The sentence at line 25 is understandable only after reading the entire paper. Please provide more understandable explanations.*

A: We changed this sentence and no longer mention single-station detections.

*R: Introduction: Moreover, what could it happen in the case of occurrence of other debris flows in the neighbouring areas? The proposed methodology could be able to identify the exact location of debris flow? The authors should also consider this eventuality in the introduction and conclusions: just write some sentences that clarify this aspect.*

A: A seismic signal of such a remote debris flow is clearly distinguishable by its amplitude decay throughout our network. We now explain this when discussing false detections.

*R: Page 4: debris flow initiation: [...] I suggest the authors to adapt the description above in the explanation of debris flow occurrence at Illgraben.*

A: We now discuss debris flow initiation with appropriate references in the manuscript.

*R: Seismic data: I suggest to eliminate IGB8 and renumbering the following seismometers.*

A: We prefer leaving the station numbering as is to keep consistency with other publications on this data set.

*R: Seismic data: Panel A of Figure 5 shows that the amplitude of signal corresponding to the green bar is very large for IGB07 while this does not appear in panel C. What about the difference between normalized ground velocity and scaled ground velocity? Some explanations in the text is due.*

A: Thanks for pointing this out. We extended the figure and its caption (see equivalent comment by Referee #1).

*R: Detection and location scheme: Points, 2, 3 and 4 at pages 6 and 7 look like statements rather than demands. They, together the explanations points to points below, could be presented at the beginning justifying the proposed approach.*

A: We ended up working these statements into the introduction of the manuscript.

*R: Detection and location scheme: The writer does not understand the computation of debris flow location through decay fitting. The analysis of the measured signal amplitude shows the exact moment of debris flow occurrence due to the high increase of the measured signal amplitude. About equation (1) Ai is a data and r is the unknown quantity. How AO can be determined? Moreover, some more explanations on the matching between RMS distribution and eq. (1) predictions could help the reader.*

A: To better explain these points, we included an additional paragraph on the location calculation and grid searching and modified the existing text.

*R: Detection and location scheme: Equation (3): What is it fit? The RMS? This should explained because most of readers are not expert on the analysis and use of seismic data.*

A: The new paragraph explains this.

*R: Results: Seismic noise ... Figures 6 and 7. The upper green triangle seems IGB10 rather than IGB9. Moreover, I suggest to label the black triangles corresponding to IGB01, IGB02 and IGB03.*

A: Sorry for the mislabeling. We corrected this.

*R: Results: Seismic noise ... Figure 6 caption. What about black cross? I do not see them.*

A: We made the black cross more obvious.

*R: Results: Seismic noise ... Line 31 of page 9. The writer does not understand the distance from variance reduction maximum: in the caption of Figure 9 there is no information about distance from variance reduction maximum as the ordinate of the panel B. Moreover, add the label 1000 and 3000 in the vertical axis.*

A: We specified what distance means and added labels accordingly.

*R: Discussion: detectability and background noise and Conclusion: Please add some comment about the influence of the distance of the seismmometer station from the debris flow occurrence location. Panel C of Figure 5 shows that only signal from IGB01, IGB02 and IGB03 are marked. Were also the other signal from IGB04-IGB10 used for computing decay fitting? In the case that station IGB01, IGB02 and IGB03 are missing the results from seismic data are the same? Please add some comment explanation.*

A: All stations were used for the ASL location. The seismograms in the mentioned Panel C simply are dominated by IGB01, IGB02 and IGB03. Including and discussing another panel in Figure 5, which shows the debris flow seismograms at a later time compared to Panel C should make this clear now.

[revised manuscript text omitted]

---

## Referee Report (RR1)

**RE:** NHESS 2016 321R1         Walter et al. Testing Seismic Amplitude Source Location (ASL) for Rapid Debris at Illgraben, Switzerland

**Overview**

This reviewed paper is quasi-ready for publication. Main deficiencies below:

1) The part of the seismic signal analysis and computation should be better and wider explained (see specific comments).

2) The references belonging to groups (see line 7 of page 4 and others) should be listed in chronological order.

3) There are a lot of oversight errors: the writer suggests a careful rereading of the work.

**Specific comments**

Title

Perhaps "Fast" is better than "Rapid"

Introduction

At line 10 of page 3 (and in other points) Burtin et al. (2013) is cited but it is not present in the references.

3 Seismic data

At line 15 of page 7, please substitute "and" with "while".

4 Detection and Location Scheme

The writer suggests some more explanations about the use of equation (3). The quantity VR is computed for each pixel of the grid where data is A(i) and fit is the second member of equation (1) computed by varying $A_O$ and $\alpha$ in the range 500-1500 times the……. and 0.0-0.001 respectively ($r_i$ is the distance between the pixel and the seismometer location). The summation, $\sum$ is carried out for the n seismometers. Did the writer understand the computation of VR?

Moreover, the writer suggests to anticipate that the computed source location could be not in the channel (see the following section at lines 24-25 of page 11) and the position of debris flow along the channel is determined by projecting the locations on the channel (as written at following section, line 11 of page 11).

5 Results: Seismic Noise....

About the sentence at lines 18-19 of page 11, the reason of bias could be due to the arrival of other solid-liquid waves?

5 Debris Flow Detection....

Why this section and the previous one have the same numbering (5)?

At line 10 of page 16, it should be Figure 13b instead of Figure 10.

6 Discussion: Suitability for early warning

The discussion about the influence of rainfall on the triggering mechanism is not clear and confused. The sentence at line 4 of page 18 "These observations could be explained…" is not reasonably linked to the previous period. Authors should write something like: "notwithstanding the two rainfalls are comparable because of nearly the same maximum intensity overo 10 minutes (data……), occurred phenomena are different because in the case of 13$^{th}$ of July, previous precipitation (please write the total depth and interval time, as for the that previous the 19$^{th}$ of July event). Then, they can add the explanation of the effect of previous precipitation.

Figures

Figure 1: please reduce the size of the triangles.

Figure 5: plot of panel c of Figure 4 and of panel b of Figure 5 should be the same but they do not seem. Moreover, the ordinate of panels a and b of Figure 5 is missing.

Figure 10: panels a and b are inverted in the caption.

---

## Referee Report (RR2)

Andrew Lockhart

March 20, 2017

This work is a retrospective test of debris flow detection at a Swiss catchment (Illgraben) using time-averaged seismic amplitudes of data archived from a temporary seismic network. The study tests this technique on a small debris flow, and compares the results against data collected from permanent instrumentation at Illgraben.  In addition, the technique was run on a 10-day period during which there were no debris flow, to check for robustness and false alarms.

Since Illgraben already has an automatic real-time debris flow monitoring system, the goal of the study is to determine if the 'seismic amplitude source location (ASL)' might provide advantages over that current system.

As applied by the authors at Illgraben, the technique is comparable to those used by other studies of debris flow detection, as well as volcanic tremor and pyroclastic flow detection and location at volcanoes. Debris flow detection is important to populations in alpine and volcanic regions. The technique has been shown to have promise but has not been fully developed (at least to my knowledge). The small number of good case studies make this one worthy of publication.

Technical queries and observations:

Body wave vs surface wave: the findings in Burtin et al that body wave dispersion fit the data better (if I understand that finding correctly) seem curious, given that debris flows are inherently a surface phenomenon.  I look forward to your future work exploring the body wave vs surface wave question.

P10, L 20 – Since Table 2 shows that your Guralp 6TD's have a frequency response from 1-100 Hz, I'm puzzled by your selection of a butterworth filter edge at 0.5 Hz. In order to keep your analysis in the linear part of the seismometer response curve, shouldn't you filter it above 1Hz? Maybe 2Hz?

P11, L11-22, and Figure 10 – This 10-minute detection time lag at CD24 with respect to the geophone detection is interesting and important. 10 minutes is a long time for this example.  Is there a volumetric component to the discrepancy? Is the ASL method detecting something like a volumetric centroid of the flow? If so, and if the flow front is just an un-energetic watery flow it is still a useful detection for hazard mitigation.

Following this theme, if a public alarm is to be disseminated as soon as possible, one would surely use the detection at or around CD1, where the discrepancy between the two systems is not great.

Figure 10, Section 5.3 – It seems possible that the lack of success using ASL to detect short-lived rock falls using short time windows may not be a good reason to avoid them for debris flows. For example, Kumagai et al (2009) used a 5-second window. Why the difference? It appears that your results using a 30-second window were similar to the 100-second window. Did you try a shorter window? Can you discuss that a little more?

Minor editorial suggestions:

P1,L15 – change 'torrents in short time.' to 'torrents in a short time.'

P7, L15 – change 'only few stations of the seismic network' to 'only a few stations of the seismic network'

P8, L15-20, and Figure 6- it's not clear but it looks like a typo may have been made (station IGB3?).
IGB09 does not have a very high amplitude, despite P8, L16.

---

## Author Response (AR2)

Dear Editor,

We thank the referees for the careful read of our manuscript and the constructive comments. As detailed below, the manuscript was modified accordingly. Furthermore, we corrected typographical mistakes and added missing panel labels in several figures.

Fabian Walter
(on behalf of the author team)

POINT-BY-POINT REPLY

Referee #1

*Referee:  Title: Perhaps "Fast" is better than "Rapid"*

Authors: Changed.

*R: Introduction: At line 10 of page 3 (and in other points) Burtin et al. (2013) is cited but it is not present in the references.*

A: The citation year in the text was wrong and should have been 2014. We corrected this.

*R: 3 Seismic data: At line 15 of page 7, please substitute "and" with "while".*

A: It seems that "while" is inappropriate. We changed "and" to a semicolon and hope this makes the text clearer.

*R: 4 Detection and Location Scheme The writer suggests some more explanations about the use of equation (3). The quantity VR is computed for each pixel of the grid where data is A(i) and fit is the second member of equation (1) computed by varying AO and α in the range 500-1500 times the……. and 0.0-0.001 respectively (ri is the distance between the pixel and the seismometer location). The summation, Σ is carried out for the n seismometers. Did the writer understand the computation of VR?*

A: Yes, this is correct. We added these explanations.

*R: Moreover, the writer suggests to anticipate that the computed source location could be not in the channel (see the following section at lines 24-25 of page 11) and the position of debris flow along the channel is determined by projecting the locations on the channel (as written at following section, line 11 of page 11).*

A: Done.

*R: 5 Results: Seismic Noise....: About the sentence at lines 18-19 of page 11, the reason of bias could be due to the arrival of other solid-liquid waves?*

A: We feel that this should be clear from the following sentence and prefer not to change this part.

5 Debris Flow Detection....

*R: Why this section and the previous one have the same numbering (5)?*

A: Our mistake, we changed the numbering.

*R: At line 10 of page 16, it should be Figure 13b instead of Figure 10.*

A: Changed.

*R: 6 Discussion: Suitability for early warning: The discussion about the influence of rainfall on the triggering mechanism is not clear and confused. The sentence at line 4 of page 18 "These observations could be explained…" is not reasonably linked to the previous period. Authors should write something like: "notwithstanding the two rainfalls are comparable because of nearly the same maximum intensity overo 10 minutes (data……), occurred phenomena are different because in the case of 13th of July, previous precipitation (please write the total depth and interval time, as for the that previous the 19th of July event). Then, they can add the explanation of the effect of previous precipitation.*

A: We changed the text accordingly. Unfortunately, we do not have access to the entire interval time.

Figures

*R: Figure 1: please reduce the size of the triangles.*

A: Done.

*R: Figure 5: plot of panel c of Figure 4 and of panel b of Figure 5 should be the same but they do not seem. Moreover, the ordinate of panels a and b of Figure 5 is missing.*

A: Thank you for the careful read. We added the missing axes labels ("Scaled vertical ground velocity (arb. u.)"). The time series in the respective panels of Figures 4 and 5 were filtered differently and are therefore not identical. We clarified this in both figure captions.

*R: Figure 10: panels a and b are inverted in the caption.*

A: Corrected (thank you for catching this).

Referee #2

*Referee 2: P10, L 20 – Since Table 2 shows that your Guralp 6TD's have a frequency response from 1-100 Hz, I'm puzzled by your selection of a butterworth filter edge at 0.5 Hz. In order to keep your analysis in the linear part of the seismometer response curve, shouldn't you filter it above 1Hz? Maybe 2Hz?*

A: The idea was to pick up some of the low-frequency signal, which is below the sensor's flat response, but which is more robust for our purposes, because it is less prone to attenuation.

*P11, L11-22, and Figure 10 – This 10-minute detection time lag at CD24 with respect to the geophone detection is interesting and important. 10 minutes is a long time for this example. Is there a volumetric component to the discrepancy? Is the ASL method detecting something like a volumetric centroid of the flow? If so, and if the flow front is just an un-energetic watery flow it is still a useful detection for hazard mitigation.*

*Following this theme, if a public alarm is to be disseminated as soon as possible, one would surely use the detection at or around CD1, where the discrepancy between the two systems is not great.*

A: Thank you for these suggestions, which make the point more clearly. We changed the text at this part and in the later early warning discussion accordingly.

*R: Figure 10, Section 5.3 – It seems possible that the lack of success using ASL to detect short-lived rock falls using short time windows may not be a good reason to avoid them for debris flows. For example, Kumagai et al (2009) used a 5-second window. Why the difference? It appears that your results using a 30-second window were similar to the 100-second window. Did you try a shorter window? Can you discuss that a little more?*

A: This is a very good point, which we had overlooked. At this point, we can only speculate that this has to do with the pulse-like character of the rock-fall signals. A brief discussion is now included.

*R: P8, L15-20, and Figure 6- it's not clear but it looks like a typo may have been made (station IGB3?). IGB09 does not have a very high amplitude, despite P8, L16.*

A: IGB09 does have high amplitude, however, this is not apparent in Figure 6a, because amplitudes are normalized. We now point this out in the text.